# WHICH EIGENVECTORS DO GRAPH TRANSFORMERS NEED FOR NODE CLASSIFICATION?

## ABSTRACT

Graph transformers have emerged as powerful tools for modeling complex graph-structured data, offering the ability to capture long-range dependencies beyond the graph adjacency. Yet their performance on node classification often lags behind that of message passing and spectral graph networks. Unlike these methods, graph transformers require additional mechanisms to inject structural information. In this work, we focus on Laplacian positional encodings, which use eigenvectors of the graph Laplacian to provide node-level positional information. Existing methods select eigenvectors using data-agnostic heuristics, assuming one-size-fits-all rules suffice. In contrast, we show that the spectral distribution of class information is graph-specific. To address this, we introduce *Broaden the Spectrum* (BTS), a novel, intuitive, and data-driven algorithm for selecting subsets of Laplacian eigenvectors for node classification. Our method is grounded in theory: we characterize the structure of optimal attention matrices for classification and show, in a simplified setting, how BTS naturally emerges as the eigenvector selection rule for achieving such attention matrices. When evaluated with standard graph transformer architectures, it delivers substantial performance gains across a wide range of node classification benchmarks. Our work shows that the performance of graph transformers on node classification has been held back by the choice of positional encodings and can be improved by employing a broader, well-chosen set of Laplacian eigenvectors.

## 1 INTRODUCTION

Graph transformers provide a flexible framework for modeling graph-structured data, with global receptive fields that can capture interactions beyond the reach of local message passing (Hoang et al., 2024). This flexibility has made them appealing for graph-level tasks such as molecular property prediction and long-range dependency modeling, where message-passing Graph Neural Networks (GNNs) often struggle with phenomena like oversquashing and oversmoothing (Topping et al., 2022; Rusch et al., 2023). Yet their performance on node classification has often lagged behind both message-passing and spectral methods (Luo et al., 2024b; Bo et al., 2023).

A central reason for this lack of performance in node-classification tasks lies in how transformers incorporate information about the graph topology. Unlike message-passing networks, which propagate information directly along edges, transformers rely on additional mechanisms to encode structure, such as positional encodings (PEs) or structural attention biases. Laplacian positional encodings (LPEs) (Dwivedi & Bresson, 2020) were among the earliest approaches for graph transformers, but structural attention biases became more prominent because they delivered stronger empirical performance on many benchmarks (Hoang et al., 2024). We argue that these performance gaps arose not from inherent limitations of LPEs, but from the simplistic eigenvector-selection heuristics used in early implementations. By replacing this heuristic with a principled selection strategy, we show that LPE-based transformers can be significantly more effective.

Most existing models truncate the Laplacian spectrum to a few lowest-frequency components (see Table 7 in Appendix C). While this low-frequency truncation heuristic works well for some homophilic graphs, it does not account for the fact that class-relevant information can appear in very different parts of the spectrum depending on the dataset. Heuristics such as fixed low-high splits in Kim et al. (2022) suggest that including a broader spectrum can help, but their effectiveness varies on the nature of the graphs themselves.

Crucially, these heuristics are data-agnostic. In contrast, the spectral GNN literature has demonstrated the advantages of *adaptive* frequency selection, combining high- and low-frequency operations based on the task (Sun et al., 2022; Bo et al., 2021; Dong et al., 2021). This motivates the need for adaptive methods that select graph transformer positional encodings in a task-aware manner, rather than relying on one-size-fits-all rules.

To address this, we introduce *Broaden the Spectrum (BTS)*, a simple, data-driven, and theoretically grounded approach for selecting Laplacian eigenvectors in graph transformers. BTS identifies the parts of the spectrum that are most aligned with class information and uses them as positional encodings. Empirically, BTS yields consistent gains across homophilic, heterophilic, and long-range benchmarks. On challenging heterophilic datasets such as Chameleon and Squirrel, even a simple transformer backbone equipped with BTS improves accuracy by more than 20%. More advanced models such as NAGphormer (Chen et al., 2023) and GraphGPS (Rampášek et al., 2022) also see substantial boosts when augmented with BTS, revealing the existence of performance bottlenecks due to under-utilization of graph topology.

**Our contributions are as follows:**

- We introduce *Broaden the Spectrum (BTS)*, a lightweight, architecture-agnostic, and data-driven algorithm for selecting Laplacian eigenvectors as positional encodings specifically for node classification.

- We provide a theoretical analysis of attention-based node classification in an illustrative linear model, showing that the optimal attention matrix has a class-wise block structure. We also derive the eigenvector selection criteria to achieve such optimal attention matrices.

- We demonstrate that including a broad and well-chosen spectrum of eigenvectors leads to significant gains in node classification performance of *existing* graph transformers across a wide range of benchmarks.

## 2 METHOD

Transformers are inherently permutation-invariant, which makes positional information essential to break symmetry and provide meaningful structure to the model. For graphs, Laplacian position encodings (LPE) have been identified by prior work (Dwivedi & Bresson, 2020; Hoang et al., 2024) to be effective, and are a natural extension of the Fourier basis used in other sequence modeling domains (Vaswani et al., 2017; Dosovitskiy et al., 2021; Nie et al., 2023). More formally, we define the LPE as follows.

**Definition 2.1** (Laplacian Position Encodings). Let $\mathcal{G} = (\mathcal{V}, \mathcal{E})$ be an undirected graph with $|\mathcal{V}| = n$ nodes, adjacency matrix $A$, and degree matrix $D$. The normalized graph Laplacian is $L = I - D^{-1/2}AD^{-1/2}$. Let $L = V\Lambda V^{\top}$ be its eigendecomposition with eigenvalues ordered as $\lambda_1 \leq \cdots \leq \lambda_n$. The *Laplacian positional encodings* are defined as $X_{\text{pos}} \in \mathbb{R}^{n \times k}$ formed by selecting any $k$ Laplacian eigenvectors.

### 2.1 CLASS-LABEL ENERGY SPECTRAL DENSITY

When using LPEs, there is a critical design choice to be made: which subset of eigenvectors should be used? The prevailing practice is to use fixed heuristics, such as using the first $k$ eigenvectors (Dwivedi & Bresson, 2020; Chen et al., 2023; Rampášek et al., 2022; Kreuzer et al., 2021; Hoang et al., 2024), or using an equal number of low- and high-frequency[1] components (Kim et al., 2022). However, Spectral GNN literature has shown the benefit of adaptively performing high- and low-frequency operations (Sun et al., 2022; Bo et al., 2021; Dong et al., 2021). Motivated by these works, we have developed a theoretically grounded method for adaptively selecting frequency components in a data-driven manner. Intuitively, our method involves finding eigenvectors that are most aligned with the class labels, which we characterize in terms of *energy spectral density*, defined as follows:

---

[1]Eigenvectors associated with small eigenvalues vary slowly across the graph, capturing "low-frequency" global variations, while eigenvectors corresponding to large eigenvalues vary rapidly, encoding "high-frequency" signals that change significantly across neighboring nodes (Shuman et al., 2013; Ortega et al., 2018).

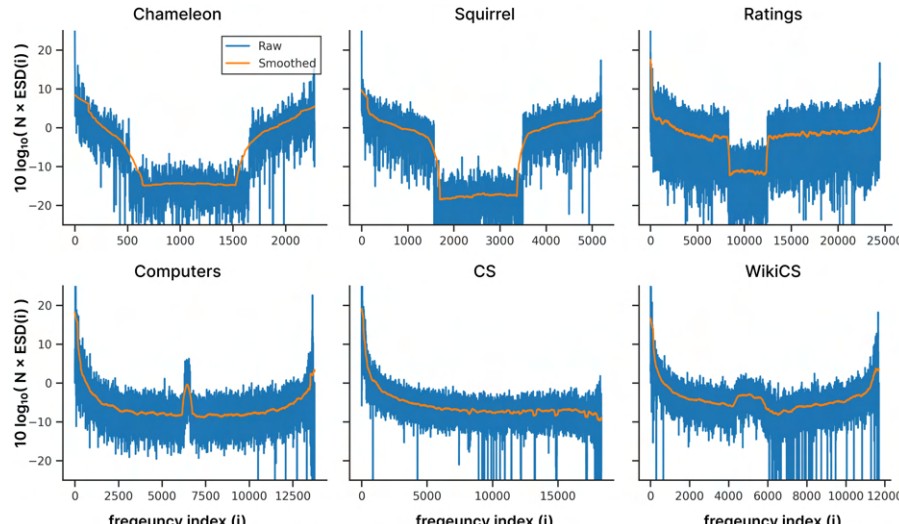

Figure 1: **Energy spectral density (ESD) of the class labels across the Laplacian spectrum for real-world graphs.** Peaks in mid- and high-frequency regions indicate that class-relevant signals are not confined to the low end of the spectrum.

**Definition 2.2** (Energy Spectral Density (ESD) of class-labels). Given the orthonormal Laplacian eigenvectors $V \in \mathbb{R}^{n \times n}$, a one-hot class-indicator matrix $Y \in \{0, 1\}^{n \times c}$, and let $V_i$ denote the $i^{\text{th}}$ column of $V$. We define the class label ESD of the class-labels:

$$\text{ESD}_i = \frac{\|V_i^\top Y\|_2^2}{\sum_{j=1}^n \|V_j^\top Y\|_2^2}, \quad i = 1, \dots, n$$

Here, $\text{ESD}_i$ measures the proportion of label energy aligned with eigenvector $V_i$. In Section 3, we show using our theoretical framework that choosing the eigenvectors with the highest ESD leads to desirable attention matrices with a class-aligned structure. Thus we rank eigenvectors by their ESD, providing a data-driven approach for efficiently utilizing the graph spectrum as positional encodings.

**ESD reveals limitations of data-agnostic selection heuristics.** Figure 1 reveals that label energy can appear in different regions of the spectrum: sometimes concentrated at low frequencies, sometimes at high frequencies, and often distributed heterogeneously. A key limitation of current eigenvector selection strategies is that they are data-agnostic, relying on fixed rules such as truncating to the lowest modes, enforcing symmetric low–high splits, or sampling at random. Because no single band is universally optimal for all graphs, such heuristics yield positional encodings that are misaligned with the task, limiting the effectiveness of graph transformers.

## 2.2 BROADEN THE SPECTRUM (BTS)

To overcome this limitation, we introduce *Broaden the Spectrum (BTS)*, a principled algorithm for selecting eigenvectors that are most informative for node classification. Rather than assuming that useful signal lies in a particular band of frequencies, we measure the alignment between eigenvectors and class labels, and select those with the highest *label ESD*. This simple yet powerful idea reframes positional encodings as a *learned spectral alignment problem*, bridging the gap between graph signal processing and transformer-based architectures.

Given a budget of $k$ eigenvectors, BTS selects the top-$k$ eigenvectors with the highest label ESD. This broadens the usable spectrum to include whichever eigenvectors carry discriminative signal, rather than assuming they reside at the bottom/top of the spectrum. Importantly, this selection is computed only from training labels, and we employ boxcar smoothing to mitigate noise. (Algorithm 1)

---

**Algorithm 1** Pseudocode for BTS eigenvector selection

---

**Input:** Laplacian eigenvectors $V \in \mathbb{R}^{n \times n}$,
    Training node indices $\mathcal{I}_{\text{train}}$, training labels $Y_{\text{train}} \in \{0,1\}^{n_{\text{train}} \times c}$,
    Number of eigenvectors to choose $k$, Smoothing window size $w$

**Output:** Indices $\mathcal{I}_k$ of selected eigenvectors

1: $V_{\text{train}} \leftarrow V[\mathcal{I}_{\text{train}}]$              ▷ Restrict to training nodes
2: $\tilde{Y}_{\text{train}} \leftarrow V_{\text{train}}^{\top} Y_{\text{train}}$             ▷ Graph Fourier transform
3: **for** $i = 1$ to $N$ **do**            ▷ Compute energy spectrum
4:   $E_i \leftarrow \|\tilde{Y}_{\text{train},i}\|_2^2$
5: $\text{ESD} \leftarrow E / \sum_{i=1}^{N} E_i$              ▷ Normalize energy
6: $\overline{\text{ESD}} \leftarrow \text{BoxcarSmooth}(\text{ESD}, w)$      ▷ Smooth with window-size $w$
7: **return** $\text{TopK}(\overline{\text{ESD}}, k)$            ▷ Return top-$k$ ESD indices

---

### 2.3 ARCHITECTURAL MODIFICATIONS

When expanding the spectrum of LPEs used in transformer models, we found it critical to incorporate a slight modification to the input encoder design. The selected eigenvectors $X_{\text{pos}}$, along with the node features $X_{\text{node}}$ are passed through two independent MLPs, followed by concatenation to form the transformer's input tokens:

$$Y_{\text{GT}^*} = \text{Transformer}\left([\text{MLP}_n(X_{\text{node}}); \text{MLP}_p(\text{norm}(X_{\text{pos}}))]\right) W_{\text{out}} \tag{1}$$

Here, $\text{norm}(\cdot)$ denotes row-wise $\ell_2$ normalization. As we show in Section 4.3, this simple modification significantly improves performance when we increase the amount of selected eigenvectors. Normalization is critical here because the scale of Laplacian eigenvector elements is $\sim 1/n$, which quickly vanishes for reasonably sized graphs. Meanwhile, independent MLPs provide a more expressive mapping from raw inputs to transformer-compatible tokens.

## 3 THEORETICAL ANALYSIS

Having introduced BTS, we now present its theoretical foundations. Positional encodings play a crucial role in shaping the attention matrices of a transformer. This motivates us to break the problem of finding a good LPE subset into two steps: (i) we first show, through an illustrative linear model, that the optimal attention matrix for node classification has a *class-wise block structure* (Section 3.1), and (ii) we find that the Laplacian eigenvectors needed to be able to approximate such a block structure are exactly the ones as described by our label-ESD-based criteria. (Section 3.2)

### 3.1 UNDERSTANDING OPTIMAL ATTENTION MATRICES FOR NODE CLASSIFICATION

Given a data matrix $X \in \mathbb{R}^{n \times d}$, the attention operation is defined as:

$$\text{Attn}(X) = \text{softmax}(X W_Q^{\top} W_K X^{\top}) X W_V^{\top}, \tag{2}$$

for some learnable weights $W_Q$, $W_K$, and $W_V$. Here, the attention score matrix, $\text{softmax}(X W_Q W_K^{\top} X)$, is constrained by the softmax operator and the dependence on $X$. We lift these constraints, simplifying the softmax-attention operation into a linear one, and ask:

> **Q1.** What form should a general attention matrix $A \in \mathbb{R}^{n \times n}$ take so that the resulting latents $Z := AX$ are most easily classifiable?

We assume a single-layer setting with a linear classifier. Given $c$ classes, let $Y \in \{0,1\}^{n \times c}$ be the one-hot class assignment matrix with $Y_{ij} = 1$ if node $i$ belongs to class $j$ and zero otherwise. The classification objective is:

$$\mathcal{L}_{\text{class}}(A, W_C) = \text{CrossEntropy}(A X W_C^{\top} \mid Y), \tag{3}$$

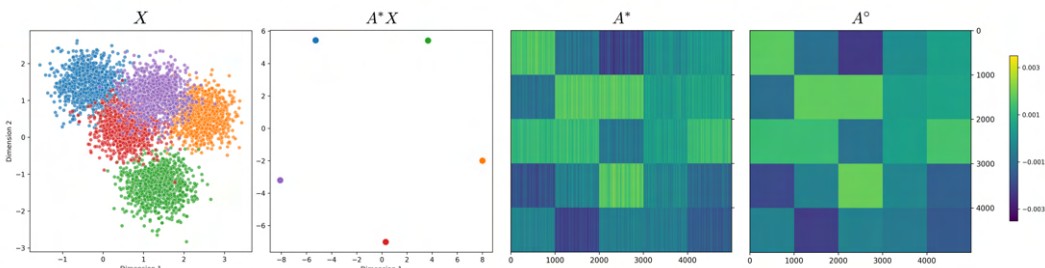

Figure 2: **Optimal Attention Matrices for Node Classification**. We find the minimizer for $\mathcal{L}_{\text{class}}$ (Equation (3)) on 2 dimensional Guassian mixture model data with 5 classes. The resulting latents and attention matrix are plotted. Upon arranging the nodes based on their class index, we observe that the corresponding optimal attention matrix, $A^*$ has an approximate class-wise block-structure of a form as predicted in $A^\circ$.

where $W_C \in \mathbb{R}^{c \times d}$ are classifier weights. We assume a mixture model $X = Y M_X + \sigma N$ for class-means $M_X \in \mathbb{R}^{c \times d}$, noise variance $\sigma > 0$, and isotropic zero-mean noise $N$ with $\mathbb{E}[NN^\top] = dI_n$ We also assume balanced classes.

We first simplify the loss formulation through the following structural lemma:

**Lemma 3.1.** *There exists a global minimizer $(A^*, W_C^*)$ of Equation (3) such that all samples from the same class are mapped to the same latent, i.e.*

$$A^* X = Y M_Z \text{ for some } M_Z \in \mathbb{R}^{c \times d}.$$

Intuitively, Lemma 3.1 (see Appendix D.1 for proof) shows that the optimal attention $A^*$ clusters latents by class. Our goal is to probe the structure of such an $A^*$, and therefore, we study the surrogate objective:

$$A^\circ = \operatorname*{argmin}_A \ \mathbb{E}_N \left[ \|AX - YM_Z\|_F^2 \right], \tag{4}$$

which leads to the following:

**Theorem 3.1** (Optimal attention has **class-wise block structure**). *Suppose $X = Y M_X + \sigma N$ as above with balanced classes. Then every minimizer $A^\circ$ of Equation (4) admits the representation*

$$A^\circ = Y M_A^\circ Y^\top, \quad \text{for some } M_A^\circ \in \mathbb{R}^{c \times c}.$$

That is, $A_{ij}^\circ$ depends only on class memberships of nodes $i$ and $j$. Theorem 3.1 (proof in Appendix D.2) formalizes the intuition that the *best attention matrix acts as a block matrix over classes*, ignoring within-class differences. This aligns with intuition; an attention matrix that is optimal for classification must satisfy two complementary objectives: (1) embeddings of nodes within the same class should be pulled tightly together, increasing the numerator of the cross-entropy softmax; and (2) embeddings of different classes should be pushed apart, reducing the denominator of the softmax. The class-wise block-diagonal components serve the first objective by encouraging within-class clustering. Whereas, the non-zero off-diagonal blocks help position clusters farther from each other in the embedding space. Simulations confirm that empirical minimizers of Equation (3) indeed exhibit this block structure (Figure 2).

### 3.2 Eigenvector Subsets for Approximating Block Attention

We now connect the block-structured optimal attention to spectral encodings. Consider a simplified linear attention formulation using only positional encodings:

$$A = X_{\text{pos}} W X_{\text{pos}}^\top \tag{5}$$

where $W$ is a learnable full-rank matrix. Our next goal is to understand:

**Q2.** Which eigenvectors allow the best approximation of the block-structured optimum $A^\circ = YMY^\top$, uniformly for any $M \in \mathbb{R}^{c \times c}$?

Essentially, our goal is to find a $k$-sized subset of eigenvectors so as to best approximate $A \approx YMY^\top$ uniformly for any $M \in \mathbb{R}^{c \times c}$. The following theorem gives a sufficient criterion as a corollary:

**Theorem 3.2** (Uniform error bound for attention approximation). *Let $V$ denote the Laplacian eigenvectors and $H \in \{0,1\}^{n \times k}$ denote an eigenvector selection matrix s.t. $H_{ij} = 1$ iff eigenvector $i$ is selected at position $j$. Define the diagonal 0/1 projector $\tilde{H} := HH^\top$, with $\tilde{H}_{ii} = 1$ iff eigenvector $i$ was selected. Set $X_{pos} = VH$ in Equation (5). Then the uniform (in $M$) error functional:*

$$\Phi(\tilde{H}) := \sup_{\|M\|_2 \leq 1} \min_W \left\| X_{pos} W X_{pos}^\top - YMY^\top \right\|_F$$

*is bounded by the residual $E := (I - \tilde{H})V^\top Y$:*

$$\Phi(\tilde{H}) \leq 2\sqrt{n}\|E\|_F + \|E\|_F^2.$$

Theorem 3.2 (see Appendix D.3 for proof) shows that the quality of approximation is controlled entirely by the residual $\|E\|_F$, i.e., how well the selected eigenvectors capture the class-indicator matrix $Y$. In other words, the smaller the projection error of $Y$, the closer the resulting attention matrix is to the block-structured optimum. This is exactly what is minimized by the class-label ESD based ranking described in our method Section 2.2, leading to the following corollary.

**Corollary 3.2.1** (Class-label ESD based eigenvector selection). *Among all $k$-sized eigenvector subsets, the selector that chooses the $k$ eigenvectors with the largest label spectral-energy $\|V_i^\top Y\|_2^2$ minimizes the bound in Theorem 3.2.*

Corollary 3.2.1 (see Appendix D.4 for proof) tells us that our label-ESD based eigenvector selection criteria would lead to class-wise block-structured attention matrices, which we have identified to be desirable in Section 3.1. Moreover, in Section 4.2, we empirically validate this prediction and show that models trained with BTS produce attention matrices with stronger class-wise block structure.

## 4 RESULTS

In this section, we evaluate our approach on a diverse set of node classification benchmarks, analyzing its effectiveness across three established graph transformer architectures. Our evaluation focuses on measuring improvements in classification performance and understanding how spectral information is utilized.

**Experimental setup.** We evaluate our approach on homophilic, heterophilic, and long-range datasets (see Appendix F for details), using three standard graph transformer backbones: GT, NAGphormer, and GraphGPS. GT (Dwivedi & Bresson, 2020) is a direct application of transformers to graphs. NAGphormer (Chen et al., 2023) restricts attention to $K$-hop neighborhoods using a normalized adjacency matrix. GraphGPS (Rampášek et al., 2022) combines message passing with transformer-based global attention and uses LapPE for positional encoding.

We use the subscript BTS to denote models tuned and trained with our ESD-based eigenvector selection approach (Section 2), as well as the input encoder modifications (Section 2.3), with $k \leq 8192$.[2] Models without the subscript follow standard truncation to the $k \leq 16$ lowest eigenvectors, (Table 7). For fairness, we also expand the GT baseline to broad spectrum setting (Section 4.3), and show that expansion alone provides little benefit without encoder modifications. Full training and hyperparameter details are given in Appendix G.

### 4.1 MAIN RESULTS

**Results on heterophilic benchmarks.** We find substantial improvements when using BTS on heterophilic graphs (Table 1). For example, on *Chameleon*, performance improves by over 22%,

---

[2]For graphs larger than 8192 nodes, we only compute the low and high 4096 eigenvectors.

Table 1: **Node classification performance on heterophilic benchmarks.** Baseline models for which results were reproduced by us are marked by †. Performance for other baselines are reported from existing literature, with "-" indicating absence of a particular evaluation in existing literature. The **top-1$^{st}$**, **top-2$^{nd}$**, and **top-3$^{rd}$** results are highlighted.

| Model | Chameleon | Squirrel | Chameleon (filt.) | Squirrel (filt.) | Tolokers | Ratings | Avg. Rank |
|---|---|---|---|---|---|---|---|
| | Accuracy ↑ | Accuracy ↑ | Accuracy ↑ | Accuracy ↑ | AU-ROC ↑ | Accuracy ↑ | ↓ |
| GCN | 38.44 ± 1.92 | 31.52 ± 0.71 | 40.89 ± 4.12 | 39.47 ± 1.47 | 83.64 ± 0.67 | 48.70 ± 0.63 | 12.50 |
| GraphSAGE | 58.73 ± 1.68 | 41.61 ± 0.74 | 37.77 ± 4.14 | 36.09 ± 1.99 | 82.43 ± 0.44 | 53.63 ± 0.39 | 9.67 |
| GAT | 48.36 ± 1.58 | 36.77 ± 1.68 | 39.21 ± 3.08 | 35.62 ± 2.06 | 83.70 ± 0.47 | 49.09 ± 0.63 | 12.00 |
| NodeFormer | 36.38 ± 3.85 | 38.89 ± 2.67 | 43.73 ± 3.26 | 37.07 ± 9.16 | 78.10 ± 1.03 | 43.79 ± 0.57 | 12.83 |
| SGFormer | 45.21 ± 3.72 | 42.65 ± 4.21 | 44.21 ± 3.06 | 43.74 ± 43.74 | - | 54.14 ± 0.62 | 6.20 |
| Exphormer | - | - | - | - | 83.53 ± 0.28 | 50.48 ± 0.34 | - |
| SpExphormer | - | - | - | - | 83.34 ± 0.31 | 50.48 ± 0.34 | - |
| Polyformer† | 63.75 ± 1.52 | 43.19 ± 2.18 | 45.49 ± 3.35 | 42.72 ± 2.25 | 85.11 ± 0.84 | 50.02 ± 0.54 | 5.83 |
| Polynormer† | 74.34 ± 1.98 | 66.91 ± 2.31 | 43.53 ± 3.20 | 42.71 ± 2.23 | 84.52 ± 0.29 | 52.72 ± 0.54 | 4.33 |
| Cobformer† | 53.82 ± 1.51 | 38.36 ± 1.81 | 45.48 ± 1.52 | 40.14 ± 1.01 | 81.85 ± 0.24 | 50.83 ± 0.59 | 8.50 |
| Specformer† | 73.49 ± 1.87 | 63.65 ± 1.88 | 40.57 ± 3.63 | 39.65 ± 1.62 | 80.86 ± 0.85 | 51.54 ± 0.28 | 7.83 |
| GT† | 50.48 ± 2.08 | 34.70 ± 1.77 | 38.07 ± 3.23 | 39.86 ± 1.96 | 80.30 ± 0.91 | 49.02 ± 0.61 | 12.50 |
| **GT$_{BTS}$** | 73.09 ± 1.00 **+22.61**↑ | 65.06 ± 1.93 **+30.36**↑ | 45.51 ± 4.69 **+7.44**↑ | 40.32 ± 1.63 **+0.46**↑ | 84.45 ± 0.66 **+4.15**↑ | 50.37 ± 0.48 **+1.35**↑ | 5.17 |
| NAGphormer† | 52.41 ± 2.21 | 40.21 ± 1.77 | 44.13 ± 3.96 | 41.82 ± 1.70 | 83.69 ± 0.86 | 50.16 ± 0.69 | 8.67 |
| **NAGphormer$_{BTS}$** | 73.90 ± 1.68 **+21.49**↑ | 65.04 ± 1.69 **+24.83**↑ | 49.01 ± 4.04 **+4.88**↑ | 43.24 ± 2.83 **+1.42**↑ | 85.47 ± 0.72 **+1.78**↑ | 49.65 ± 0.65 **-0.51**↓ | 4.00 |
| GraphGPS† | 60.92 ± 2.54 | 43.43 ± 1.46 | 44.86 ± 3.72 | 43.16 ± 2.10 | 86.29 ± 0.68 | 50.19 ± 0.51 | 5.33 |
| **GraphGPS$_{BTS}$** | 73.16 ± 1.70 **+12.24**↑ | 65.87 ± 1.30 **+22.44**↑ | 44.26 ± 3.99 **-0.60**↓ | 44.74 ± 2.10 **+1.58**↑ | 86.31 ± 0.63 **+0.02**↑ | 51.33 ± 0.58 **+1.14**↑ | 3.17 |

Table 2: **Node classification accuracy (%) on homophilic benchmarks.** Baseline models for which results were reproduced by us are marked by †. Performance for other baselines are reported from existing literature, with "-" indicating absence of a particular evaluation in existing literature. The **top-1$^{st}$**, **top-2$^{nd}$**, and **top-3$^{rd}$** results are highlighted.

| Model | Physics | CS | Photo | Computers | WikiCS | ogbn-arXiv | Avg. Rank |
|---|---|---|---|---|---|---|---|
| | Accuracy ↑ | Accuracy ↑ | Accuracy ↑ | Accuracy ↑ | Accuracy ↑ | Accuracy ↑ | ↓ |
| NodeFormer | 96.45 ± 0.28 | 95.64 ± 0.22 | 93.46 ± 0.35 | 86.98 ± 0.62 | 74.73 ± 0.94 | 59.90 ± 0.42 | 11.33 |
| SGFormer | 96.60 ± 0.18 | 94.78 ± 0.34 | 95.10 ± 0.47 | 91.99 ± 0.70 | 73.46 ± 0.56 | 72.63 ± 0.13 | 8.67 |
| Exphormer | 96.89 ± 0.09 | 94.93 ± 0.01 | 95.35 ± 0.22 | 91.47 ± 0.17 | 78.19 ± 0.29 | 71.27 ± 0.27 | 8.33 |
| SpExphormer | 96.70 ± 0.05 | 95.00 ± 0.15 | 95.33 ± 0.44 | 91.09 ± 0.08 | 78.20 ± 0.14 | 70.82 ± 0.24 | 9.33 |
| Polyformer | 97.06 ± 0.24 | 95.57 ± 0.31 | 95.98 ± 0.51 | 92.22 ± 0.45 | 79.69 ± 0.45 | 71.73 ± 0.28 | 3.00 |
| Polynormer | 96.62 ± 0.23 | 95.28 ± 0.37 | 95.58 ± 0.61 | 91.74 ± 0.64 | 79.16 ± 0.68 | 71.23 ± 0.27 | 7.00 |
| CoBformer | 96.49 ± 0.08 | 94.99 ± 0.14 | 94.12 ± 0.33 | 90.87 ± 0.27 | 80.26 ± 0.21 | 72.69 ± 0.12 | 8.00 |
| Specformer | 97.04 ± 0.20 | 95.84 ± 0.40 | 92.18 ± 0.62 | 92.87 ± 0.30 | 80.23 ± 0.64 | 70.20 ± 0.15 | 5.33 |
| GT | 96.02 ± 0.20 | 94.66 ± 0.44 | 91.59 ± 0.68 | 85.65 ± 0.59 | 72.91 ± 0.59 | 55.68 ± 0.39 | 14.00 |
| **GT$_{BTS}$** | 96.90 ± 0.18 **+0.88**↑ | 95.44 ± 0.33 **+0.78**↑ | 95.95 ± 0.48 **+4.36**↑ | 91.46 ± 0.51 **+5.81**↑ | 78.94 ± 0.26 **+6.03**↑ | 70.30 ± 0.12 **+14.62**↑ | 6.83 |
| NAGphormer | 96.98 ± 0.13 | 95.71 ± 0.26 | 95.51 ± 0.41 | 91.39 ± 0.41 | 78.73 ± 0.66 | 69.43 ± 0.32 | 7.33 |
| **NAGphormer$_{BTS}$** | 97.05 ± 0.18 **+0.07**↑ | 95.42 ± 0.39 **-0.29**↓ | 95.90 ± 0.37 **+0.39**↑ | 91.85 ± 0.44 **+0.46**↑ | 79.42 ± 0.55 **+0.69**↑ | 71.29 ± 0.13 **+1.86**↑ | 4.83 |
| GraphGPS | 97.13 ± 0.17 | 95.70 ± 0.38 | 95.35 ± 0.45 | 91.64 ± 0.46 | 77.67 ± 0.73 | 65.16 ± 1.45 | 7.17 |
| **GraphGPS$_{BTS}$** | 97.21 ± 0.14 **+0.08**↑ | 95.72 ± 0.37 **+0.02**↑ | 95.87 ± 0.42 **+0.52**↑ | 91.87 ± 0.45 **+0.23**↑ | 79.47 ± 0.48 **+1.80**↑ | 70.92 ± 0.33 **+5.76**↑ | 3.67 |

and on *Squirrel*, by over 30% when BTS-filtered eigenvectors are used with a simple transformer architecture (GT). Remarkably, this brings the vanilla transformer architecture into close competition with, and in some cases even surpassing, more complex graph transformer models proposed in recent literature. To the best of our knowledge, the performance reported here for *Chameleon*, *Squirrel*, and *Tolokers* represents the strongest results achieved by any graph transformer model to date.

These improvements can be understood through the lens of the class-label ESD. As shown in Figure 1, graphs like *Chameleon* and *Squirrel* exhibit significant class energy in a broad set of low and high frequency regions. These findings highlight that the historical reliance on low-frequency truncation was a critical bottleneck, masking the true representational and generalization potential of graph transformers.

**Results on homophilic benchmarks.** As shown in Table 2, BTS improves performance even on homophilic graphs. GT$_{BTS}$ achieves +5.8% on *Computers*, +6.0% on *WikiCS*, and +14.4% on *ogbn-arXiv*. Gains for NAGphormer and GraphGPS are smaller but consistent. These results are expectedly more modest than in heterophilic settings, as the spectral energy of class signals in many homophilic graphs is concentrated in low frequencies. Still, BTS captures the broader spectrum whenever useful, yielding robust improvements.

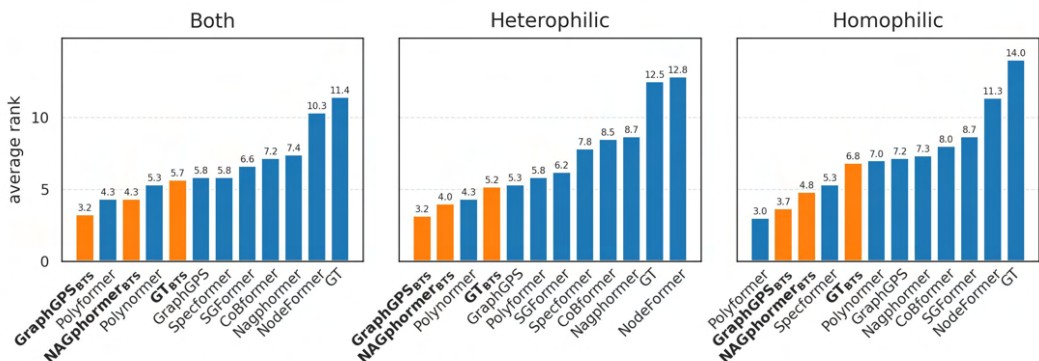

Figure 3: **Average rank of models across datasets.** Simple architectures become competitive with recent models upon incorporating BTS.

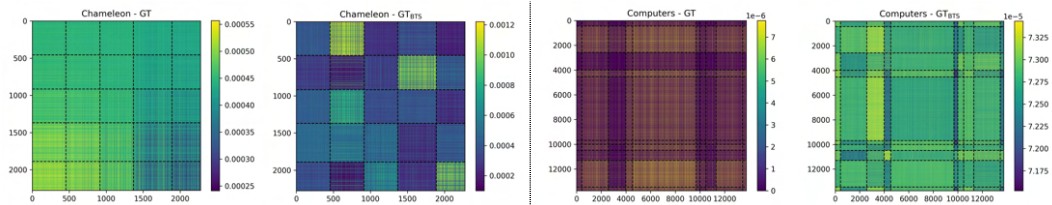

Figure 4: **Attention matrices learned by vanilla GT and GT$_{\text{BTS}}$ on Chameleon (left) and Computers (right).** Incorporating BTS leads to clearer class-wise block structures, consistent with our theoretical prediction that such attention patterns are optimal for node classification (Section 3.1). These attention matrices correspond to the first head in the last layer after softmax, and nodes are sorted according to their classes. Full results from all layers and heads can be found in Appendix E.

**Results on long-range benchmarks.** Graph transformers are naturally suited for capturing long-range dependencies due to their global attention mechanism. However, a number of recent papers have shown that graph transformers suffer from overglobalization (Xing et al., 2024) and perform poorly on long-range tasks. As shown in Table 3, our method achieves substantial improvements on the long-range benchmark (Liang et al., 2025) over baseline graph transformer architectures. These datasets exhibit particularly strong gains when using BTS-selected features. For instance, performance for GT on *Paris* improves by over 38%, and on *Shanghai* by 31%, bringing the vanilla transformer model on par with strong baseline methods.

Table 3: **Node classification accuracy (%) on Long Range Benchmarks** The **top-1st**, **top-2nd**, and **top-3rd** results are highlighted.

| Model | Paris | Shanghai |
|---|---|---|
| GCN | $47.30 \pm 0.20$ | $52.40 \pm 0.30$ |
| GraphSAGE | $49.10 \pm 0.60$ | $60.40 \pm 0.30$ |
| SGFormer | $45.00 \pm 0.20$ | $53.5 \pm 0.30$ |
| GT | $15.46 \pm 3.93$ | $21.05 \pm 0.51$ |
| **GT$_{\text{BTS}}$** | $53.79 \pm 0.17$ | $52.66 \pm 0.83$ |
| $\Delta$ | **+38.33 ↑** | **+31.61 ↑** |
| NAGphormer | $25.26 \pm 0.34$ | $24.94 \pm 0.28$ |
| **NAGphormer$_{\text{BTS}}$** | $53.68 \pm 0.23$ | $57.26 \pm 0.34$ |
| $\Delta$ | **+28.42 ↑** | **+32.32 ↑** |
| GraphGPS | $28.99 \pm 0.31$ | $28.46 \pm 0.31$ |
| **GraphGPS$_{\text{BTS}}$** | $54.12 \pm 0.23$ | $55.31 \pm 0.33$ |
| $\Delta$ | **+25.13 ↑** | **+26.85 ↑** |

## 4.2 ATTENTION MATRICES

Our theoretical analysis (Section 3) predicts that the optimal attention matrix for node classification should have a class-wise block structure and that using BTS-selected eigenvectors would lead to such attention matrices. To validate this, we examine the attention matrices obtained after training. As shown in Figure 4, models trained with BTS yield attention matrices that indeed display a substantially stronger block-wise organization aligned with class partitions. Since BTS models perform better, this also suggests that class-wise block structures are indeed desirable attention patterns. See Section E for attention matrices across all layers and heads of the networks.

### 4.3 ABLATIONS

We conduct ablation studies to isolate the contributions of two key factors: (i) the design of the encoder that processes positional encodings, and (ii) the strategy used to select eigenvectors. Our results show that architectural support is necessary to benefit from larger $k$, while principled selection is essential for avoiding overfitting and consistently improving performance.

**Spectral expansion and encoder design.** The results in Table 4 indicate that expanding $k$ (num. of eigenvectors) beyond this range yields only marginal improvements without our input encoder modifications. For example, with GT, Chameleon's performance increases from 50.48% to 52.28%. This partly explains why most approaches have restricted positional encodings to a small set of low-frequency eigenvectors. However, we see substantial gains when the positional encodings are normalized and processed with a MLP. With these modifications, performance improves to 67.83% on Chameleon, and from 34.70% to 62.91% on Squirrel. This indicates that, while a broader spectrum of eigenvectors provides useful signal, careful design of the encoder is needed to fully utilize them.

Table 4: **Ablation of our encoder modifications.** Increasing number of eigenvectors ($k$) beyond low frequencies alone offers only marginal gains without our architectural modifications.

| Position Encoder | k | Chameleon | Squirrel | WikiCS | Computers |
|---|---|---|---|---|---|
| Baseline (linear) | $k \leq 16$ | $50.48 \pm 2.08$ | $34.70 \pm 1.77$ | $72.91 \pm 0.59$ | $85.65 \pm 0.59$ |
| Baseline (linear) | no-bound | $52.28 \pm 2.88$ | $37.71 \pm 1.79$ | $73.75 \pm 0.63$ | $87.21 \pm 0.55$ |
| Norm + Linear | no-bound | $65.07 \pm 1.08$ | $56.96 \pm 1.23$ | $77.67 \pm 0.48$ | $91.39 \pm 0.40$ |
| Norm + MLP | no-bound | $\mathbf{67.83} \pm \mathbf{1.82}$ | $\mathbf{62.91} \pm \mathbf{1.04}$ | $\mathbf{78.46} \pm \mathbf{0.56}$ | $\mathbf{91.66} \pm \mathbf{0.41}$ |

**Eigenvector selection strategies.** Next, we explore how different eigenvector selection heuristics influence performance. Table 5 shows that simply using the full spectrum severely hurts generalization due to overfitting, demonstrating that indiscriminate inclusion of all frequencies is detrimental. We also tested four different selection heuristics: low-only, high-only, low+high, and low+medium+high, each corresponding to retaining different bands of the spectrum. While all four of these heuristics show improvements over baseline performance, we find that their effectiveness is not consistent across datasets. For example, the best out of the four variants is high-only on both heterophilic datasets (Chameleon, Squirrel) and low-only on homophilic datasets (WikiCS, Computers). In contrast, our data-driven selection (BTS) consistently achieves the best results.

Table 5: Impact of different eigenvector selection strategies on the performance of GT*. The **top-1[st]**, **top-2[nd]**, and **top-3[rd]** results are highlighted.

| Eigenvector Selection | Chameleon | Squirrel | WikiCS | Computers |
|---|---|---|---|---|
| Baseline | $50.48 \pm 2.08$ | $34.70 \pm 1.77$ | $72.91 \pm 0.59$ | $85.65 \pm 0.59$ |
| Full spectrum | $48.14 \pm 3.05$ | $35.30 \pm 1.61$ | $72.35 \pm 0.72$ | $85.20 \pm 0.30$ |
| Low-only | $67.83 \pm 1.82$ | $62.91 \pm 1.04$ | $\mathbf{78.46} \pm \mathbf{0.56}$ | $\mathbf{91.26} \pm \mathbf{0.41}$ |
| High-only | $\mathbf{72.79} \pm \mathbf{1.37}$ | $\mathbf{63.22} \pm \mathbf{1.75}$ | $73.45 \pm 0.40$ | $89.80 \pm 0.55$ |
| Low + High (equal) | $72.50 \pm 0.93$ | $63.19 \pm 1.49$ | $78.37 \pm 0.52$ | $90.99 \pm 0.50$ |
| Low + Medium + High (equal) | $66.51 \pm 1.88$ | $44.66 \pm 1.59$ | $75.38 \pm 0.56$ | $89.79 \pm 0.33$ |
| **BTS** | $\mathbf{73.09} \pm \mathbf{1.68}$ | $\mathbf{65.06} \pm \mathbf{1.93}$ | $\mathbf{78.94} \pm \mathbf{0.26}$ | $\mathbf{91.46} \pm \mathbf{0.51}$ |

## 5 RELATED WORK

**Graph Transformers.** Graph Transformers (GTs) allow graph nodes to interact globally via self-attention (Dwivedi & Bresson, 2020). Because self-attention is naturally permutation-equivariant, additional mechanisms are needed to encode graph structure, such as positional encodings (PEs) or structural attention biases. PEs work by adding or concatenating a *position vector* (or encoding) to the node-level tokens before they are processed by the transformer (Dwivedi & Bresson, 2020; Dwivedi et al., 2022; Kreuzer et al., 2021; Chen et al., 2023). Whereas methods using structural attention biases work by directly manipulating the attention matrices and biasing the node-node interactions (Ying et al., 2021; Airale et al., 2025; Li et al., 2025; Park et al., 2022; Luo et al., 2024a; Rampášek et al., 2022; Stoll et al., 2025; Kong et al., 2023; Zhao et al., 2021). Laplacian positional encodings

(LPEs) (Dwivedi & Bresson, 2020) were among the earliest approaches for graph transformers, and later structural attention biases became more prominent.

**Graph Transformers and Positional Encodings.** Within the positional encoding approach Laplacian eigenvectors have become a common choice (Hoang et al., 2024), providing a spectral basis that reflects graph topology. However, nearly all existing GTs adopt a heuristic truncation: retaining only the first $k$ low-frequency eigenvectors. The low-frequency bias is evident in models such as GT (Dwivedi & Bresson, 2020), NAGphormer (Chen et al., 2023), GraphTrans (Wu et al., 2021), GraphGPS (Rampášek et al., 2022), UGT (Lee et al., 2024), SAN (Kreuzer et al., 2021), and SAT (Chen et al., 2022). Scalability-oriented variants such as ANS-GT (Zhang et al., 2022), Gapformer (Liu et al., 2023), and Exphormer (Shirzad et al., 2023), which focus on efficient attention also retain the same positional encoding heuristic. TokenGT (Kreuzer et al., 2021) extends beyond low frequencies by including both low- and high-frequency eigenvectors, but relies on fixed splits rather than task-driven selection.

**The Role of High-Frequency Signals in Node Classification.** In contrast, the MPNN literature has increasingly emphasized the importance of high-frequency information for node classification, especially in heterophilic graphs. Spectral methods explicitly operate in the frequency domain and modulate both low- and high-frequency signals (Wu et al., 2019; Dong et al., 2020). Early spectral models such as Spectral CNN (Bruna et al., 2013) and ChebNet (Defferrard et al., 2016) introduced learnable filters over eigenvalues, and subsequent works demonstrated that high-frequency information is critical for node-level expressiveness, such as adaptive propagation (Chien et al., 2020), complete spectral filtering (Luan et al., 2020), and frequency-based feature selection (Bo et al., 2021). These techniques explicitly exploit high-frequency modes, while adaptive approaches such as AdaGNN (Dong et al., 2021), and spectral attention (Chang et al., 2021) dynamically balance contributions from different parts of the spectrum.

## 6 CONCLUSION

In this paper, we introduced Broaden the Spectrum (BTS), a lightweight and architecture-agnostic algorithm for selecting Laplacian eigenvectors as positional encodings. We show that spectral distribution of class information is unique for each graph, making fixed heuristics for eigenvector selection (such as low-frequency truncation) inherently limited. BTS addresses this by selecting eigenvectors in a data-driven way, consistently improving the performance of graph transformers across homophilic, heterophilic, and long-range benchmarks. In several cases, BTS elevates even simple transformer architectures to state-of-the-art levels.

To ground our method, we also developed a theoretical framework for attention-based node classification, showing that the optimal attention matrix exhibits a class-wise block structure and that eigenvectors most correlated with labels are best suited to approximate it. While we focus on the Laplacian spectrum, our framework is general and can be extended to other orthonormal bases. However, both BTS and our theoretical analysis are tailored to the supervised node classification setting, and it remains an open direction to adapt similar principles to other tasks such as link prediction, graph classification, or self-supervised pretraining.

Overall, our work highlights that the performance bottleneck of graph transformers on node classification lies in how positional encodings are chosen. By broadening the spectrum with a data-driven, theoretically grounded selection, we show that this bottleneck can be overcome and that even simple graph transformers are competitive and reach near state-of-the-art performance.

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

APPENDIX

# A    ABLATIONS FOR ALL ARCHITECTURES

We conducted an ablation study in Section 4.3 to evaluate the impact of higher-order spectral components, encoder design, and label-aware selection on the Graph Transformer (GT). Here, we extend this analysis to all baseline transformer architectures. As shown in Table 6, the same trends hold across models, showing that access to more eigenvectors and a better encoder design can improve performance.

Table 6: Node classification performance with full eigenvector spectrum vs with left-truncated spectrum but tuned $K$.

| | | Heterophilic | | Homophilic | |
| --- | --- | --- | --- | --- | --- |
| **Model** | **Eigenvectors** | **Chameleon** | **Squirrel** | **WikiCS** | **Computers** |
| GT | $K \in [4, 16]$ (tuned) | $50.48 \pm 2.08$ | $34.70 \pm 1.77$ | $72.91 \pm 0.59$ | $85.65 \pm 0.59$ |
| GT | $K \in [4, N]$ (tuned) | $52.28 \pm 2.88$ | $37.71 \pm 1.79$ | $73.75 \pm 0.63$ | $87.21 \pm 0.55$ |
| GT* Linear | $K \in [4, N]$ (tuned) | $65.07 \pm 1.08$ | $56.96 \pm 1.23$ | $77.67 \pm 0.48$ | $91.39 \pm 0.40$ |
| GT* MLP | $K \in [4, N]$ (tuned) | $67.83 \pm 1.82$ | $62.91 \pm 1.04$ | $78.46 \pm 0.56$ | $91.66 \pm 0.41$ |
| GT* | full spectrum (fixed) | $48.14 \pm 3.05$ | $35.30 \pm 1.61$ | $72.35 \pm 0.72$ | $85.20 \pm 0.30$ |
| GT$_{BTS}$ | BTS, $K \in [4, N]$ (tuned) | $73.09 \pm 1.68$ | $65.06 \pm 1.93$ | $78.94 \pm 0.26$ | $91.46 \pm 0.51$ |
| NAGphormer | $K \in [4, 16]$ (tuned) | $52.41 \pm 2.21$ | $40.21 \pm 1.77$ | $78.73 \pm 0.66$ | $91.39 \pm 0.41$ |
| NAGphormer | $K \in [4, N]$ (tuned) | $57.06 \pm 1.96$ | $40.89 \pm 2.06$ | $79.70 \pm 0.50$ | $91.61 \pm 0.42$ |
| NAGphormer* Linear | $K \in [4, N]$ (tuned) | $- \pm -$ | $- \pm -$ | $79.33 \pm 0.44$ | $92.16 \pm 0.48$ |
| NAGphormer* MLP | $K \in [4, N]$ (tuned) | $70.07 \pm 2.33$ | $63.87 \pm 1.51$ | $79.83 \pm 0.63$ | $91.96 \pm 0.37$ |
| NAGphormer* | full spectrum (fixed) | $59.63 \pm 2.06$ | $52.27 \pm 1.28$ | $79.63 \pm 0.63$ | $91.53 \pm 0.47$ |
| NAGphormer$_{BTS}$ | BTS, $K \in [4, N]$ (tuned) | $73.90 \pm 1.68$ | $65.04 \pm 1.69$ | $79.42 \pm 1.55$ | $91.85 \pm 0.44$ |
| GraphGPS | $K \in [4, 16]$ (tuned) | $60.92 \pm 2.54$ | $43.43 \pm 1.46$ | $77.67 \pm 0.73$ | $91.64 \pm 0.46$ |
| GraphGPS | $K \in [4, N]$ (tuned) | $64.67 \pm 2.98$ | $47.12 \pm 4.21$ | $77.40 \pm 0.45$ | $91.60 \pm 0.45$ |
| GraphGPS* Linear | $K \in [4, N]$ (tuned) | $68.8 \pm 2.13$ | $57.67 \pm 0.97$ | $78.92 \pm 0.53$ | $91.96 \pm 0.22$ |
| GraphGPS* MLP | $K \in [4, N]$ (tuned) | $70.24 \pm 2.08$ | $63.40 \pm 1.16$ | $78.68 \pm 0.46$ | $91.80 \pm 0.40$ |
| GraphGPS* | full spectrum (fixed) | $59.14 \pm 1.95$ | $42.88 \pm 1.77$ | $77.42 \pm 0.98$ | $91.24 \pm 0.40$ |
| GPS$_{BTS}$ | BTS, $K \in [4, N]$ (tuned) | $73.16 \pm 1.70$ | $65.87 \pm 1.30$ | $79.47 \pm 0.48$ | $91.87 \pm 0.45$ |

# B    PERFORMANCE OF BTS IN LABEL SCARCE SETTINGS

In this experiment, we investigate whether BTS remains effective when the number of available labels is severely limited. We subsample the training labels at four different ratios and evaluate GT and GT$_{BTS}$ under identical budget constraints across four datasets. As shown in Figure 5, reducing label availability lowers accuracy for both methods, which is expected in few-shot settings. Nonetheless, GT$_{BTS}$ consistently outperforms GT across all datasets and label ratios. These results demonstrate that BTS eigenvector selection criteria reliably remains higher performing as compared to naive top-$k$ truncation.

# C    HIGHEST MAXIMUM FREQUENCIES USED IN THE LITERATURE

To better understand the typical frequency truncation choices in existing graph transformer models, we compile representative values of the maximum number of Laplacian eigenvectors ($k$) used across a range of published works. As shown in Table 7, most methods restrict $k$ to a small number—often below 16—reinforcing the low-pass inductive bias observed in current practice.

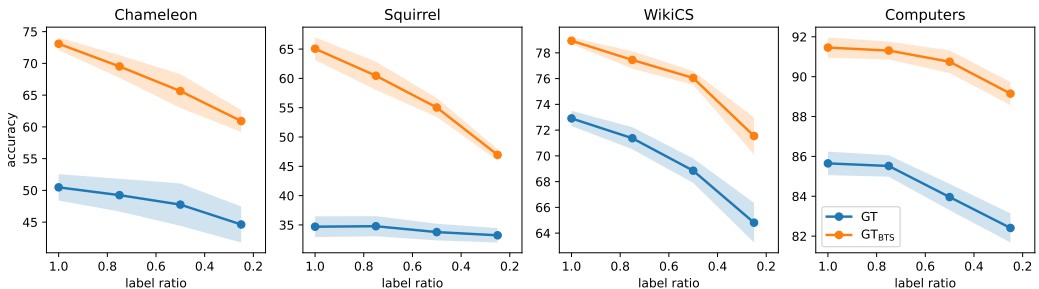

Figure 5: Comparison of GT and $GT_{BTS}$ in the label-scarce setting. $GT_{BTS}$ consistently outperforms GT despite decreasing label budgets, confirming the robustness of BTS-based eigenvector.

Table 7: Maximum number of eigenvectors ($K_{max}$) used by recent graph transformer models, based on publicly available code.

| Model | $K_{max}$ |
|---|---|
| NAGphormer (Chen et al., 2023) | 15 |
| GraphGPS (Rampášek et al., 2022) | 10 |
| SAN (Kreuzer et al., 2021) | 10 |
| GT (Dwivedi & Bresson, 2020) | 10 |
| UGT (Lee et al., 2024) | 10 |
| Exphormer (Shirzad et al., 2023) | 10 |

# D PROOFS

## D.1 PROOF FOR LEMMA 3.1

*Proof.* Given a data matrix $X \in \mathbb{R}^{n \times d}$ and the minimization problem:

$$\min_{A, W_C} \mathcal{L}_{class}(A, W_C), \qquad (6)$$

our goal is to characterize the associated latents $Z^* := A^* X$ of an optimal solution $(A^*, W_C^*)$.

Let $(A, W_C)$ be an arbitrary candidate solution to the minimization problem (6) and define the resulting latents $Z := AX$ consisting of rows $\mathbf{z}_i \in \mathbb{R}^d$. Let $Y \in \{0, 1\}^{n \times c}$ denote the one-hot class indicator matrix, $y_i$ denote the class index of node $i$, and $n_j$ is the number of points in class $j$. First we write the cross entropy classification objective:

$$\mathcal{L}_{class}(A, W_C) = \sum_{i=1}^{n} \ell\left(W_C \mathbf{z}_i, y_i\right), \qquad (7)$$

with $\ell(\mathbf{u}, y_i) = -\mathbf{u}_{y_i} + \log \sum_{j=1}^{c} e^{\mathbf{u}_j}$ for any predicted logits $\mathbf{u} \in \mathbb{R}^c$. We note that $\ell(\cdot, y_i)$ is convex in $\mathbf{u}$.

Now define the $j^{th}$-class mean-latent:

$$\overline{\mathbf{z}}_j := \frac{1}{n_j} \sum_{i: y_i = j} \mathbf{z}_i. \quad \forall j \in \{1 \ldots c\} \qquad (8)$$

By Jensen's inequality, since $\ell$ is convex and $\mathbf{z} \mapsto W_C \mathbf{z}$ is linear, we have

$$\frac{1}{n_j} \sum_{i: y_i = j} \ell\left(W_C \mathbf{z}_i, y_i\right) \geq \ell\left(W_C \left(\frac{1}{n_j} \sum_{i: y_i = j} \mathbf{z}_i\right), y_i\right) \quad \forall j \in \{1 \ldots c\}$$

$$\triangleq \ell\left(W_C \overline{\mathbf{z}}_j, y_i\right), \qquad (9)$$

i.e.

$$\sum_{i: y_i = j} \ell(W_C \mathbf{z}_i, y_i) \geq n_j \ell\left(W_C \overline{\mathbf{z}}_j, y_i\right). \quad \forall j \in \{1 \ldots c\} \qquad (10)$$

Hence, if we define $Z^*$ as the latent matrix with every row $\mathbf{z}_i^*$ set to $\bar{\mathbf{z}}_{y_i}$, then we have for any $W_C$:

$$\mathcal{L}_{\text{class}}(Z^*, W_C) \leq \mathcal{L}_{\text{class}}(Z, W_C). \tag{11}$$

In particular, if we minimize over all $W_C$, we have:

$$\min_{W_C} \mathcal{L}_{\text{class}}(Z^*, W_C) \leq \min_{W_C} \mathcal{L}_{\text{class}}(Z, W_C). \tag{12}$$

Thus, from any (approximately) optimal $Z$ we can construct a collapsed $Z^*$ (with $\mathbf{z}_i^* := \bar{\mathbf{z}}_{y_i}$) that is *at least as good* as $Z$. This relationship holds for any $W_C$, therefore after solving the minimization problem in Equation (6) we obtain a global minimizer $(A^*, W_C^*)$ with:

$$A^* X \triangleq Z^* = Y M_Z \tag{13}$$

where $M_Z \in \mathbb{R}^{c \times d}$ is the matrix of class mean-latents with rows $\bar{\mathbf{z}}_j \in \mathbb{R}^d$. $\qquad\square$

## D.2 PROOF FOR THEOREM 3.1

*Proof.* Given $X$ and $Y$ defined as in the proof from Appendix D.1, assume a mixture model for the data matrix:

$$X = Y M_X + \sigma N \tag{14}$$

where $M_X \in \mathbb{R}^{c \times d}$ is a matrix of class-means, $\sigma > 0$ denotes noise variance, and $N$ denotes isotropic zero-mean noise with $\mathbb{E}[N N^T] = d I_n$.

Consider the noise-averaged surrogate objective defined in Equation (4):

$$\min_A \mathcal{L}_{\text{attn}}(A) = \mathbb{E}_N \left[ \|AX - Y M_Z\|_F^2 \right] \tag{15}$$

where $M_Z \in \mathbb{R}^{c \times d}$ is the class mean-latent matrix as defined in Lemma 3.1 and Appendix D.1. We shall omit the $N$ subscript from $\mathbb{E}_N$ for notational convenience. The purpose of the surrogate loss is to probe the structure of an attention matrix that attempts to solves the classification problem defined in Equation (12).

Assume $Y$ is defined such that the classes are balanced, i.e. $Y^\top Y = \frac{n}{c} I_c$. Then define the orthogonal projector onto $\text{col}(Y)$:

$$P := Y(Y^\top Y)^{-1} Y^\top = \frac{c}{n} Y Y^\top \tag{16}$$

Note that $P$ is symmetric and idempotent. Our goal is to reveal class-wise block structure of attention matrices that minimize the surrogate objective.

First, let the attention $A \in \mathbb{R}^{n \times n}$ be arbitrary. Since $Y M_Z \in \text{col}(Y) = \text{range}(P)$, we have the orthogonal decomposition:

$$\|AX - Y M_Z\|_F^2 = \|P(AX - Y M_Z) + (I_n - P)(AX - Y M_Z)\|_F^2$$
$$= \|P(AX - Y M_Z)\|_F^2 + \|(I_n - P)AX\|_F^2 \tag{17}$$

Now define $\tilde{A} := PA$ and notice that, by idempotency of $P$, the first term in the decomposition stays the same while the second term vanishes. Hence, we have:

$$\left\| \tilde{A} X - Y M_Z \right\|_F^2 = \|P(AX - Y M_Z)\|_F^2 \leq \|AX - Y M_Z\|_F^2 \tag{18}$$

Thus, since the inequality is preserved under expectation, we have $\mathcal{L}_{\text{attn}}(PA) \leq \mathcal{L}_{\text{attn}}(A)$ for all $A$. In particular, given a minimizer $A^\circ$ of $\mathcal{L}_{\text{attn}}$, we have $\mathcal{L}_{\text{attn}}(PA^\circ) = \mathcal{L}_{\text{attn}}(A^\circ)$. But then by the decomposition, we know that

$$0 = \mathbb{E} \left[ \|(I_n - P)A^\circ X\|_F^2 \right] \geq \sigma^2 d \|(I_n - P)A^\circ\|_F^2 \tag{19}$$

using the data model assumption $X = Y M_X + \sigma N$. Hence, we have $PA^\circ = A^\circ$.

Similarly, for any $A$, we can again using the data model assumption to expand the surrogate loss:

$$\mathcal{L}_{\text{attn}}(A) = \|AY M_X - Y M_Z\|_F^2 + \sigma^2 d \|A\|_F^2 \tag{20}$$

By definition of $P$, since $YM_X \in \text{col}(Y)$, we have $PYM_X = YM_X$, i.e. the signal term stays the same if we make the substitution $\tilde{A} := AP$. Hence, by idempotency of $P$ (and consequently idempotency of $I_n - P$),

$$\mathcal{L}_{\text{attn}}(A) - \mathcal{L}_{\text{attn}}(AP) = \sigma^2 d \left( \|A\|_F^2 - \|AP\|_F^2 \right)$$
$$= \sigma^2 d \left( \text{tr}(AA^\top) - \text{tr}(APA^\top) \right)$$
$$= \sigma^2 d \left( \text{tr}(A(I_n - P)A^\top) \right)$$
$$= \sigma^2 d \left\| A(I_n - P) \right\|_F^2 \geq 0 \qquad (21)$$

Again, if we take a minimizer $A^\circ$ of $\mathcal{L}_{\text{attn}}$, then this tells us that $\mathcal{L}_{\text{attn}}(A^\circ) = \mathcal{L}_{\text{attn}}(A^\circ P)$, i.e. $\|A^\circ(I_n - P)\|_F = 0$, i.e. $A^\circ = A^\circ P$.

Combining the results so far, we can conclude that any minimizer $A^\circ$ admits the representation

$$A^\circ = P A^\circ P$$
$$= Y M_A^\circ Y^\top \qquad (22)$$

with $M_A^\circ := \frac{c^2}{n^2}(Y^\top A Y)$. Since we know a minimizer will admit such a representation, we can restrict the surrogate loss in terms of the kernel matrix $M_A$:

$$\mathcal{L}_{\text{attn}}(Y M_A Y^\top) = \mathbb{E}\left[ \left\| Y M_A Y^\top (Y M_X + \sigma N) - Y M_Z \right\|_F^2 \right]$$
$$= \left\| Y \left( \frac{n}{c} M_A M_X - M_Z \right) \right\|_F^2 + \sigma^2 \mathbb{E}\left[ \left\| Y M_A Y^\top N \right\|_F^2 \right]$$
$$= \frac{n}{c} \left\| \frac{n}{c} M_A M_X - M_Z \right\|_F^2 + \frac{\sigma^2 n^2 d}{c^2} \|M_A\|_F^2 \qquad (23)$$

Lastly, we show that $M_A^\circ := \frac{c}{n} M_Z M_X^\top (M_X M_X^\top + \frac{c}{n} \sigma^2 d I_c)^{-1}$ yields a closed-form minimizer for the surrogate loss, and it is in fact the *unique* minimizer. To do so, consider another candidate solution specified by $M_A := M_A^\circ + \Delta$. We expand the loss:

$$\mathcal{L}_{\text{attn}}\left( Y M_A Y^\top \right) = \frac{n}{c} \left\| \frac{n}{c} M_A^\circ M_X - M_Z + \frac{n}{c} \Delta M_X \right\|_F^2 + \frac{\sigma^2 n^2 d}{c^2} \|M_A^\circ + \Delta\|_F^2$$
$$= \mathcal{L}_{\text{attn}}\left( Y M_A^\circ Y^\top \right) + \frac{n}{c} \left\| \frac{n}{c} \Delta M_X \right\|_F^2 + \frac{\sigma^2 n^2 d}{c^2} \|\Delta\|_F^2 + 2\mathcal{R}(\Delta) \qquad (24)$$

where $\mathcal{R}(\Delta)$ is given by:

$$\mathcal{R}(\Delta) = \frac{n}{c} \left\langle \frac{n}{c} \Delta M_X, \frac{n}{c} M_A^\circ M_X - M_Z \right\rangle_F + \frac{\sigma^2 n^2 d}{c^2} \langle \Delta, M_A^\circ \rangle_F$$
$$= \frac{n^2}{c^2} \left[ \left\langle \Delta, \left( \frac{n}{c} M_A^\circ M_X - M_Z \right) M_X^\top + \sigma^2 d M_A^\circ \right\rangle_F \right] \qquad (25)$$

We can expand the right term in the Frobenius inner product using the proposed solution $M_A^\circ$:

$$\left( \frac{n}{c} M_A^\circ M_X - M_Z \right) M_X^\top + \sigma^2 d M_A^\circ$$
$$= M_Z M_X^\top \left( M_X M_X^\top + \frac{c}{n} \sigma^2 d I_c \right)^{-1} M_X M_X^\top + \frac{c}{n} \sigma^2 d M_Z M_X^\top \left( M_X M_X^\top + \frac{c}{n} \sigma^2 d I_c \right)^{-1} - M_Z M_X^\top$$
$$= M_Z M_X^\top \left( M_X M_X^\top + \frac{c}{n} \sigma^2 d I_c \right)^{-1} \left( M_X M_X^\top + \frac{c}{n} \sigma^2 d I_c \right) - M_X M_X^\top$$
$$= M_Z(M_X^\top - M_X^\top)$$
$$= 0 \qquad (26)$$

Hence, $\mathcal{R}(\Delta) = 0$, i.e. $\mathcal{L}_{\text{attn}}(Y M_A Y^\top) \leq \mathcal{L}_{\text{attn}}(Y M_A^\circ Y^\top)$ with equality iff $\Delta = 0$. Indeed, $M_A^\circ$ is the unique solution. $\qquad \square$

### D.3 PROOF FOR THEOREM 3.2

*Proof.* To reiterate the setup, we consider the simplified linear attention formulation using only positional encodings, as in Equation (5). Let $V$ denote the Laplacian eigenvectors and $H \in \{0,1\}^{n \times k}$ denote an eigenvector selection matrix s.t. $H_{ij} = 1$ iff eigenvector $i$ is selected at position $j$. Define the diagonal 0/1 projector $\tilde{H} := HH^\top$, with $\tilde{H}_{ii} = 1$ iff eigenvector $i$ was selected, and set $X_{\text{pos}} := VH$ in the formulation. Then define the error functional over all $M \in \mathbb{R}^{c \times c}$ within the unit ball $\|M\|_2 \le 1$:

$$\Phi(\tilde{H}) := \sup_{\|M\|_2 \le 1} \min_W \left\| X_{\text{pos}} W X_{\text{pos}}^\top - YMY^\top \right\|_F$$

$$= \sup_{\|M\|_2 \le 1} \min_W \|VHWH^\top V^\top - YMY^\top\|_F. \tag{27}$$

The error functional defines the quality of an approximation of the block-structured representation $YMY^\top$, uniformly across all kernels $M$. Our goal is to provide a uniform error bound for this functional.

We first observe that since eigenvectors are chosen without replacement, $VH$ will have full column rank. Hence, the inner optimization problem is a well-defined classical linear least squares problem that can be solved exactly for $W$:

$$W^*(\tilde{H}) := H^\top V^\top YMY^\top VH. \tag{28}$$

Hence, the functional becomes:

$$\Phi(\tilde{H}) = \sup_{\|M\|_2 \le 1} \|VHH^\top V^\top YMY^\top VHH^\top V^\top - YMY^\top\|_F$$

$$= \sup_{\|M\|_2 \le 1} \|\tilde{H}V^\top YMY^\top V\tilde{H} - V^\top YMY^\top V\|_F$$

$$= \sup_{\|M\|_2 \le 1} \|\tilde{H}V^\top YM(\tilde{H}V^\top Y)^\top - V^\top YMY^\top V\|_F \tag{29}$$

since $\tilde{H}$ is symmetric. Define the residual $E := (\mathbb{I}_n - \tilde{H})V^\top Y$. Then we can rewrite the inner norm and bound it:

$$\|\tilde{H}V^\top YM(\tilde{H}V^\top Y)^\top - V^\top YMY^\top V\|_F = \|(V^\top Y - E)M(V^\top Y - E)^\top - V^\top YMY^\top V\|_F$$

$$= \|EMY^\top V + V^\top YME^\top - EME^\top\|_F$$

$$\le \|M\|_2 \left(2\|E\|_F\|V^\top Y\|_F + \|E\|_F^2\right)$$

$$\le \|M\|_2 \left(2\sqrt{n}\|E\|_F + \|E\|_F^2\right) \tag{30}$$

since $V$ is orthonormal. Hence, within the unit ball $\|M\|_2 \le 1$, we have the upper bound:

$$\|\tilde{H}V^\top YM(\tilde{H}V^\top Y)^\top - V^\top YMY^\top V\|_F \le 2\sqrt{n}\|E\|_F + \|E\|_F^2. \tag{31}$$

By definition, we have:

$$\Phi(\tilde{H}) \le 2\sqrt{n}\|E\|_F + \|E\|_F^2. \tag{32}$$

Thus, $\|E\|_F$ gives us uniform control over the upper error bound of the given $\tilde{H}$ over all $\|M\|_2 \le 1$. $\qquad\square$

### D.4 PROOF FOR COROLLARY 3.2.1

*Proof.* Following the conclusion of Theorem 3.2, we showed that the uniform error bound for attention approximation is controlled by the residual $\|E\|_F$, where $E := (I - \tilde{H})V^\top Y$. To minimize $\|E\|_F$, we first observe that:

$$\|E\|_F^2 \triangleq \sum_{i=1}^n \left(1 - \tilde{H}_{ii}\right) \|V_i^\top Y\|_2^2 \tag{33}$$

where $\tilde{H}_{ii}$ indicates whether or not eigenvector $i$ was included, and $s_i := \|V_i^\top Y\|_2^2$ is the $\ell_2$-norm of the $i^{\text{th}}$ row of $V^\top Y$. Thus, to minimize $\|E\|_F$, we should include eigenvectors corresponding to the $k$ largest $s_i$'s (in terms of *spectral-energy* $\|s_i\|_2^2$), i.e. the optimal selector $\tilde{H}^*$ that minimizes the upper error bound is exactly the one based on the top $k$ indices of the $s_i$'s. $\qquad\square$

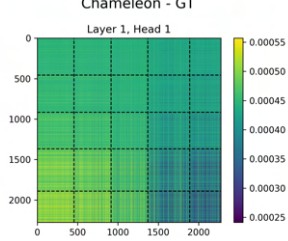

Figure 6: Attention matrix learned by vanilla GT on Chameleon. The model was selected based on best validation performance and uses a single transformer layer with one attention head.

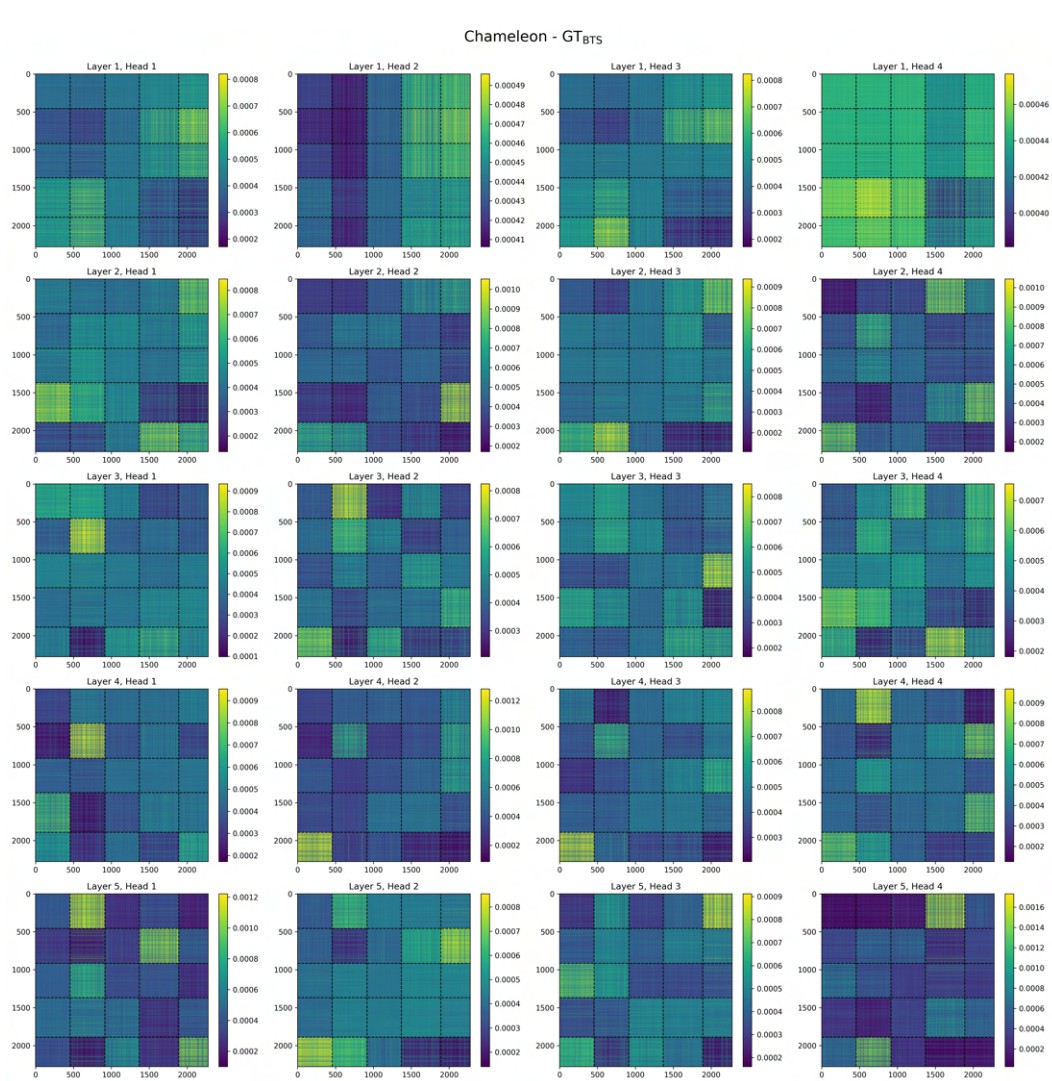

Figure 7: Attention matrices learned by GT$_{BTS}$ on Chameleon, shown for every layer and head. In practice, computations such as intra-class clustering and inter-class separation can be distributed across layers and heads.

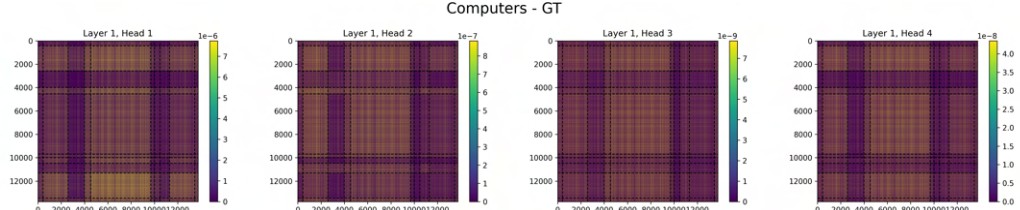

**Figure 8:** Attention matrices learned by vanilla GT on Computers, shown for every head (single-layer model). The learned attention matrices do not exhibit clear block structure and hence provide limited contribution to intra-class clustering and inter-class separation.

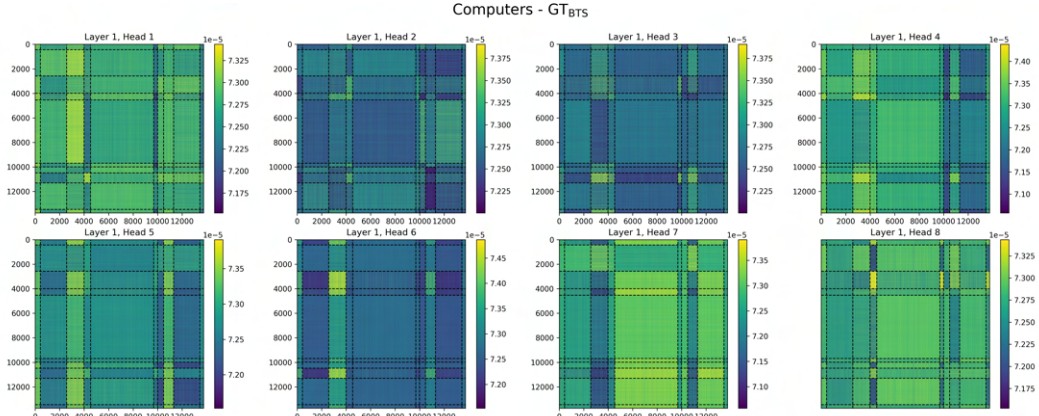

**Figure 9:** Attention matrices learned by vanilla GT on Computers, shown for every head (single-layer model). In practice, computations such as intra-class clustering and inter-class separation can be distributed across layers and heads.

# E  FULL ATTENTION MATRICES

We present learned attention matrices from *all* layers and heads in GT and GT$_{BTS}$ on Chameleon (Figures 6 and 7) and Computers (Figures 8 and 9), extending the plots in Figure 4 which only considered the first head of the last layer of each model. The number of layers and heads are different in each case as they were chosen independently through hyperparameters sweeps (which we describe in Appendix G). All models have the best validation performance in their respective hyperparameter sweeps.

From the figures, we can see that the different layers and heads in GT$_{BTS}$ variants exhibit clear block structure, and also that intra-class clustering and inter-class separation can be distributed across heads and layers. In contrast, vanilla GT typically shows much weaker block structure.

# F  BASELINE SOURCES

## F.1  HETEROPHILIC DATASETS

We use five real-world datasets with graphs that have a homophily level $\leq 0.30$: Actor (Pei et al., 2020), Chameleon and Squirrel (Rozemberczki et al., 2021), as well as Ratings and Tolokers (Platonov et al., 2023). Key statistics for these datasets are listed in Table 8. We follow the experimental setup in (Pei et al., 2020) for Actor, Chameleon, and Squirrel, and for Ratings and Tolokers, we adopt the setup described in (Platonov et al., 2023), using the 10 train/validation/test splits provided.

The results for GCN-based methods and heterophily-based methods in Table 8 for Actor, Chameleon, and Squirrel have been sourced from (Azabou et al., 2023). Similarly, results for Ratings and Tolokers are sourced from (Platonov et al., 2023), while results for transformer-based methods across all datasets are obtained from (Shirzad et al., 2024).

Table 8: Statistics of heterophilic datasets used in our experiments.

| DATASET | NODES | EDGES | CLASSES | HOMOPHILY RATIO |
|---------|-------|-------|---------|-----------------|
| CHAMELEON | 2,277 | 31,421 | 5 | 0.23 |
| SQUIRREL | 5,201 | 198,493 | 5 | 0.22 |
| TOLOKERS | 11,758 | 519,000 | 2 | 0.09 |
| RATINGS | 244,92 | 39,402 | 5 | 0.14 |

Table 9: Statistics of homophilic datasets used in our experiments.

| DATASET | NODES | EDGES | CLASSES | HOMOPHILY RATIO |
|---------|-------|-------|---------|-----------------|
| PHYSICS | 34,493 | 495,924 | 5 | 0.92 |
| CS | 18,333 | 81,894 | 15 | 0.83 |
| PHOTO | 7,650 | 238,162 | 8 | 0.84 |
| COMPUTERS | 13,752 | 491,722 | 10 | 0.79 |
| WIKICS | 11,701 | 216,123 | 10 | 0.66 |
| OGBN-ARXIV | 169,343 | 1,166,243 | 40 | 0.65 |

## F.2 HOMOPHILIC DATASETS

We use five real-world datasets: Amazon Computers and Amazon Photos (McAuley et al., 2015), Coauthor CS and Coauthor Physics (Sinha et al., 2015), and WikiCS (Mernyei & Cangea, 2022). Key statistics for these datasets are listed in Table 9. The experimental setup follows that of (Shirzad et al., 2024), where the datasets are split into development and test sets. All hyperparameter tuning is performed on the development set, and the best models are subsequently evaluated on the test set.

We use a 60:20:20 train/validation/test split for the Amazon and Coauthor datasets. The results reported for all datasets in Table 2 are sourced from (Shirzad et al., 2024).

## F.3 LONG RANGE BENCHMARK DATASETS

To evaluate the ability of models to capture long-range dependencies, we use the City-Networks benchmark (Liang et al., 2025), which consists of large-scale road network graphs derived from OpenStreetMap data. We focus on two representative cities—Paris and Shanghai—which feature grid-like topology, low clustering coefficients, and large diameters. These characteristics make them particularly well-suited for studying long-range signal propagation. Key statistics for these datasets are provided in Table 10.

Following the experimental protocol in (Liang et al., 2025), we perform transductive node classification using a 10:10:80 train/validation/test split. The node labels are defined by eccentricity-based quantiles, ensuring that the task inherently depends on information from distant nodes.

Table 10: Statistics of City-Networks datasets used in our experiments.

| DATASET | NODES | EDGES | CLASSES | HOMOPHILY RATIO |
|---------|-------|-------|---------|-----------------|
| PARIS | 114,127 | 182,511 | 10 | 0.70 |
| SHANGHAI | 183,917 | 262,092 | 10 | 0.75 |

## F.4 BASELINE MODEL PERFORMANCE ACROSS DATASETS FROM EXISTING LITERATURE

The previously reported performance of baseline models (GT, GraphGPS, and NAGphormer) on multiple graph datasets is summarized in Table 11. The reported values, sourced from existing literature.

Table 11: Performance across datasets for GT, GraphGPS, and NAGphormer models previously reported in existing literature.

| Dataset | GT | GraphGPS | NAGphormer |
|---|---|---|---|
| Chameleon | - | 40.79 ± 4.03 | - |
| Squirrel | - | 39.67 ± 2.84 | - |
| Tolokers | - | 83.71 ± 0.48 | 78.32 ± 0.95 |
| Ratings | - | 53.10 ± 0.42 | 51.26 ± 0.72 |
| Physics | 97.05 ± 0.05 | 97.12 ± 0.19 | 97.34 ± 0.03 |
| CS | 94.64 ± 0.13 | 93.93 ± 0.12 | 95.75 ± 0.09 |
| Photo | 94.74 ± 0.13 | 95.06 ± 0.13 | 95.49 ± 0.11 |
| Computers | 91.18 ± 0.17 | 91.19 ± 0.54 | 91.22 ± 0.14 |
| WikiCS | - | 78.66 ± 0.49 | 77.16 ± 0.72 |
| Arxiv | - | 70.97 ± 0.41 | 70.13 ± 0.55 |

# G  ADDITIONAL TRAINING DETAILS

**Optimizer.**   We use the AdamW optimizer (Loshchilov & Hutter, 2019) for all runs, and fixed the number of epochs to 200. We additionally employ the linear-warmup-cosine-decay learning rate schedule. Linear rate warmup happens over 10 epochs (fixed), and cosine decay happens over the remaining 190 epochs (also fixed). All other hyperparameters are chosen by the tuning algorithm explained below.

**Hyperparameter tuning.**   We optimize hyperparameters using the Tree-structured Parzen Estimator (TPE) algorithm (Bergstra et al., 2011), as implemented in Optuna (Akiba et al., 2019). The complete hyperparameter space used in our experiments is detailed in Table 12. The number of tuning trials is adjusted based on the size of each dataset: for graphs with up to 7,500 nodes, we perform 300 tuning trials; for graphs with up to 15,000 nodes, we allow 200 trials; and for larger graphs, we limit the number of trials to 100. Performing complete hyperparameter tuning, on a machine with $4 \times$ NVidia L40S GPUs takes 2-4 hours depending on the size of the dataset. Hyperparameters are selected based on validation-set performance, and all reported results correspond to test-set performance using the best configurations found. The scripts used for hyperparameter tuning are also included in our codebase.

# H  LLM USAGE DISCLOSURE

We used LLMs solely for the purpose of editing and polishing the paper.

Table 12: Complete hyperparameter search space for all model variants presented in this paper.

| Hyperparameter | Search Space | Sampling type |
|---|---|---|
| **Common parameters** | | |
| Learning rate | $[10^{-4}, 10^{-1}]$ | Logarithmic |
| Weight Decay | $[10^{-7}, 10^{-2}]$ | Logarithmic |
| Dropout | $[0, 0.5]$ | Linear |
| Attention Dropout | $[0, 0.5]$ | Linear |
| Window Length | $\{256, 512, 1024, 2048, 4096\}$ | |
| Transformer Depth | $\{1, 2, \ldots, 8\}$ | Linear |
| Number of attention heads | $\{0, 1, 2, 4, 8\}$ | |
| **Common for GT$_{BTS}$/NAGphormer$_{BTS}$/GraphGPS$_{BTS}$** | | |
| Number of eigenvectors ($K$) | $\{4, 8, 16, \ldots, 1024\}$ | |
| Pos. feature encoder - output dimension | $\{8, 16, 32, 64, 128\}$ | |
| Pos. feature encoder - hidden dimension | $\{16, 32, 64, \ldots, 2048\}$ | |
| Pos. feature encoder - # hidden layers | $\{1, 2, 3, 4\}$ | |
| Node feature encoder - output dimension | $\{8, 16, 32, 64, 128\}$ | |
| Node feature encoder - hidden dimension | $\{16, 32, 64, 128\}$ | |
| Node feature encoder - # hidden layers | $\{1\}$ | |
| **Specific for GT** | | |
| Number of eigenvectors ($K$) | $\{4, 8, 16\}$ | |
| Token dimension | $\{64, 128, 256, 512\}$ | |
| **Specific for NAGphormer** | | |
| Number of eigenvectors ($K$) | $\{4, 8, 16\}$ | |
| Token dimension | $\{64, 128, 256, 512\}$ | |
| Number of hops (same for NAGphormer$_{BTS}$) | $\{1, 2, 3, \ldots, 20\}$ | |
| **Specific for GraphGPS** | | |
| Number of eigenvectors ($K$) | $\{4, 8, 16\}$ | |
| LPE - number of layers | $\{1, 2, 3, \ldots, 8\}$ | |
| LPE - number of post-layers | $\{0, 1, 2, 3, 4\}$ | |

