# OpenReview forum: "Which Eigenvectors Do Graph Transformers Need for Node Classification?"
_ICLR.cc/2026/Conference — Submitted to ICLR 2026_

### Official Review · Reviewer_3HAs · 2025-10-17

**Soundness:** 2
**Presentation:** 3
**Contribution:** 2
**Rating:** 4
**Confidence:** 5

**Summary:**

This paper examines the positional encoding effectiveness of Laplacian matrix eigenvectors in graph transformers, with a particular focus on node classification tasks. It introduces an Energy Spectral Density metric derived from class labels and uses this metric to identify the top-_k_ eigenvectors for encoding. The proposed approach is integrated into several existing graph transformer architectures, leading to consistent improvements in node classification performance across multiple datasets.

**Strengths:**

1. The proposed ESD metric and corresponding BTS method are simple, intuitive, and easily adaptable to a wide range of graph transformer models.
2. The paper offers a theoretical analysis of the rationale behind BTS, elucidating its effectiveness in the context of node classification tasks.
3. The experimental evaluation is thorough, including extensive ablation studies that validate the efficacy of the proposed BTS method.

**Weaknesses:**

1. The first one lies in BTS’s reliance on full eigen-decomposition, which incurs higher computational complexity compared to previous methods that select only the lowest or highest top-_k_ eigenvectors. This substantially restricts its scalability to large-scale graphs.
2. Both the BTS method and its theoretical analysis are limited to node classification task, posing considerable challenges when extending to other tasks such as link prediction or graph-level prediction.
3. The assumptions underlying the theoretical analysis are overly restrictive—particularly the formulation  $X = Y M_X + \sigma N $—which does not accurately reflect real-world conditions, where node features typically incorporate structural and neighborhood information.
4. The experimental section omits several important baselines, including PolyFormer [1] and SpecFormer [2].
5. Moreover, the Chameleon and Squirrel datasets used in the experiments exhibit substantial edge overlap, results should be reported on their filtered versions in [3].
6. (Minor) The graph should be denoted as "an undirected graph" in Definition 2.1.

[1] Ma J, He M, Wei Z. Polyformer: Scalable node-wise filters via polynomial graph transformer[C]//Proceedings of the 30th ACM SIGKDD Conference on Knowledge Discovery and Data Mining. 2024: 2118-2129.

[2] Bo D, Shi C, Wang L, et al. Specformer: Spectral graph neural networks meet transformers[J]. arXiv preprint arXiv:2303.01028, 2023.

[3] Platonov O, Kuznedelev D, Diskin M, et al. A critical look at the evaluation of GNNs under heterophily: Are we really making progress?[J]. arXiv preprint arXiv:2302.11640, 2023.

**Questions:**

1. Please begin by responding to the Weaknesses part.
2. The baselines PolyFormer and SpecFormer should be included for comparison in the experimental evaluation.

---

> ### Author Response · Authors · 2025-11-21
> **Rebuttal (1/3)**
>
> We thank the reviewer for their thorough and constructive feedback. We are grateful to know that the reviewer finds our “proposed ESD metric and corresponding BTS method are simple, intuitive, and easily adaptable,” and that our “experimental evaluation is thorough.” We present a point-by-point response to the reviewer’s concerns and comments below.
>
> >**W1. The first one lies in BTS’s reliance on full eigen-decomposition, which incurs higher computational complexity compared to previous methods that select only the lowest or highest top-k eigenvectors. This substantially restricts its scalability to large-scale graphs.**
>
> We would like to clarify that BTS does **not** rely on full eigen-decomposition. In fact, for large graphs (bigger than 8192 nodes), we do not compute the full spectrum. We instead only compute the first and last 4096 eigenvectors, and then apply BTS to select the best eigenvectors *amongst those*. This is mentioned in the footnote on Page 6 of our manuscript of our paper. Importantly, our theoretical analysis and the BTS selection algorithm are fully compatible with any available subset of eigenvectors, whether one computes 100, 1000, or 10000\.
>
> It is true that our paper also highlights a broader empirical trend: selecting from a wider portion of the Laplacian spectrum improves performance. However, this trend is *not* specific to BTS. As shown in our ablations (Table 5 on page 9 of our paper), even the lowest–top-k heuristic benefits from access to more eigenvectors. What matters is that BTS consistently outperforms top-k under the same eigenvector budget.
>
> >**W2. Both the BTS method and its theoretical analysis are limited to node classification task, posing considerable challenges when extending to other tasks such as link prediction or graph-level prediction.**
>
> While it is intriguing to consider how one might select LPEs for graph-level classification or link prediction, we would like to clarify the motivation and scope of our work. We focus specifically on **node classification**, because this is the setting where graph transformers have historically struggled to match the performance of MPNNs and spectral GNNs \[1, 2\].
>
> Other tasks pose different challenges for injecting positional or structural information, and have been addressed extensively in prior work \[3, 4\]. Our contribution is therefore focused and complementary: we provide a theoretical framework and a practical solution tailored to node classification, the regime where graph transformers have been most disadvantaged. There is also strong precedent for this focus, as several prior works have studied only node classification for similar reasons \[1, 5, 6, 7, 8, 9, 10, 12, 13\].
>
> >**W3. The assumptions underlying the theoretical analysis are overly restrictive—particularly the formulation $Y \= MX \+ \\sigma N$—which does not accurately reflect real-world conditions, where node features typically incorporate structural and neighborhood information.**
>
> We acknowledge that, in our theoretical model, node features are class-separable embeddings that do not incorporate structural or neighborhood information of the graph. However, our empirical results suggest that our theoretical model is sufficiently expressive for the phenomenon we seek to understand: (1) it led directly to our eigenvector-selection method, which yields strong empirical gains across datasets, and (2) we do indeed see real attention matrices exhibit the same class-wise block structure predicted by our analysis (Section 4.2 on page 8).
>
> Our goal here was not to model the full complexity of real-world graphs, but to study a more focused question: *given node embeddings from which class information is already recoverable, what form should an optimal attention matrix take?* The scenario we analyze can be viewed as an embedding model for the later layers of a graph transformer, where representations have already begun to cluster by class and the remaining layers refine this separation. The simplified setting allowed us to derive a clean and interpretable characterization of attention matrices and isolate the impact of eigenvector selection.

---

> ### Author Response · Authors · 2025-11-21
> **Rebuttal (2/3)**
>
> >**W4. The experimental section omits several important baselines, including PolyFormer \[1\] and SpecFormer \[2\].**
>
> Thank you for your suggestion. We have added results from four more baselines: **PolyFormer** \[11\], **Specformer** \[1\], **Polynormer** \[12\], and **CoBFormer** \[13\] (the latter two requested by reviewer wJCD). Because published results were not available for all datasets, we reproduced these methods ourselves, ensuring a fair comparison under the same data splits and a similar hyperparameter-tuning setup. These new results have been added to **Tables 1 and 2 (page 7\) and Figure 3 (page 8\) of the updated manuscript** (highlighted in blue).
>
> Based on **average rank** (lower is better), we observe the following trends:
>
> - **Heterophilic datasets:** `GraphGPS-BTS < NAGphormer-BTS < Polynormer < GT-BTS < GraphGPS < …`
> - **Homophilic datasets:** `PolyFormer < GraphGPS-BTS < NAGphormer-BTS < SpecFormer < GT-BTS < …`
> - **Overall:** `GraphGPS-BTS < PolyFormer < NAGphormer-BTS < Polynormer < GT-BTS < …`
>
> GraphGPS goes from being ranked 6th overall to 1st, when equipped with BTS. Notably, GT, the simplest transformer in our study, improves dramatically, going from being ranked 12th overall to 5th. Overall, the BTS variants of GT, NAGphormer, and GraphGPS remain highly competitive with these recent state-of-the-art models. This further supports our central claim: LPE-based graph transformers were primarily held back by the choice of eigenvectors, and with a principled selection strategy, even simple architectures can achieve strong node-classification performance.
>
> **Table R4A:** Performance on heterophilic datasets
>
> | Model | Chameleon | Squirrel | Chameleon (filtered) | Squirrel (filtered) | Tolokers | Ratings | Average Rank |
> | :---: | :---: | :---: | :---: | :---: | :---: | :---: | :---: |
> | GT-BTS | 73.09 ± 1.00 | 65.06 ± 1.93 | 45.51 ± 4.69 | 40.32 ± 1.63 | 84.45 ± 0.66 | 50.37 ± 0.48 | 5.17 |
> | NAGphormer-BTS | 73.90 ± 1.68 | 65.04 ± 1.69 | 49.01 ± 4.04 | 43.24 ± 2.83 | 85.47 ± 0.72 | 49.65 ± 0.65 | 4.00 |
> | GraphGPS-BTS | 73.16 ± 1.70 | 65.87 ± 1.30 | 44.26 ± 3.99 | 44.74 ± 2.10 | 86.31 ± 0.63 | 51.33 ± 0.58 | 3.17 |
> | Polyformer | 63.75 ± 1.52 | 43.19 ± 2.18 | 45.49 ± 3.35 | 42.72 ± 2.25 | 85.11 ± 0.84 | 50.02 ± 0.54 | 5.83 |
> | Polynormer | 74.34 ± 1.98 | 66.91 ± 2.31 | 43.53 ± 3.20 | 42.71 ± 2.23 | 84.52 ± 0.29 | 52.72 ± 0.54 | 4.33 |
> | CoBformer | 53.82 ± 1.51 | 38.36 ± 1.81 | 45.48 ± 1.52 | 40.14 ± 1.01 | 81.85 ± 0.24 | 50.83 ± 0.59 | 8.50 |
> | Specformer | 73.49 ± 1.87 | 63.65 ± 1.88 | 40.57 ± 3.63 | 39.65 ± 1.62 | 80.86 ± 0.85 | 51.54 ± 0.28 | 7.83 |
>
> **Table R4B:** Performance on homophilic datasets
>
> | Model | Physics | CS | Photo | Computers | WikiCS | ogbn-arXiv | Average Rank |
> | :---: | :---: | :---: | :---: | :---: | :---: | :---: | :---: |
> | GT-BTS | 96.90 ± 0.18 | 95.44 ± 0.33 | 95.95 ± 0.48 | 91.46 ± 0.51 | 78.94 ± 0.26 | 70.30 ± 0.12 | 6.83 |
> | NAGphormer-BTS | 97.05 ± 0.18 | 95.42 ± 0.39 | 95.90 ± 0.37 | 91.85 ± 0.44 | 79.42 ± 0.55 | 71.29 ± 0.13 | 4.83 |
> | GraphGPS-BTS | 97.21 ± 0.14 | 95.72 ± 0.37 | 95.87 ± 0.42 | 91.87 ± 0.45 | 79.47 ± 0.48 | 70.92 ± 0.33 | 3.67 |
> | Polyformer | 97.06 ± 0.24 | 95.57 ± 0.31 | 95.98 ± 0.51 | 92.22 ± 0.45 | 79.69 ± 0.45 | 71.73 ± 0.28 | 3.00 |
> | Polynormer | 96.61 ± 0.23 | 95.28 ± 0.37 | 95.58 ± 0.61 | 91.74 ± 0.64 | 79.16 ± 0.68 | 71.23 ± 0.27 | 7.00 |
> | CoBformer | 96.49 ± 0.08 | 94.99 ± 0.14 | 94.12 ± 0.33 | 90.87 ± 0.27 | 80.26 ± 0.21 | 72.69 ± 0.12 | 8.00 |
> | Specformer | 97.04 ± 0.20 | 95.84 ± 0.40 | 92.18 ± 0.62 | 92.87 ± 0.30 | 80.23 ± 0.64 | 70.20 ± 0.15 | 5.33 |
>
> >**W5. Moreover, the Chameleon and Squirrel datasets used in the experiments exhibit substantial edge overlap, results should be reported on their filtered versions in \[3\].**
>
> Thank you for mentioning this\! We used the older unfiltered splits because most papers have reported on the older splits. As you requested, we have added the results for newer filtered splits in **Table 1** of our updated manuscript. For convenience, we provide a few focused results on these datasets in Tables R4A and R4B above (under W4).
>
> >**W6. (Minor) The graph should be denoted as "an undirected graph" in Definition 2.1.**
>
> Thanks for pointing this out\! We have fixed this in our updated manuscript.

---

> > ### Author Response · Authors · 2025-11-21
> > **Rebuttal (3/3)**
> >
> > **References**
> >
> > \[1\] Bo, Deyu, et al. "Specformer: Spectral Graph Neural Networks Meet Transformers." ICLR 2023\.
> >
> > \[2\] Luo, Yuankai, Lei Shi, and Xiao-Ming Wu. "Classic GNNs are Strong Baselines: Reassessing GNNs for Node Classification." NeurIPS 2024\.
> >
> > \[3\] Shomer, Harry, et al. "Lpformer: An adaptive graph transformer for link prediction." ACM SIGKDD 2024\.
> >
> > \[4\] Mialon, Grégoire, et al. "Graphit: Encoding graph structure in transformers." arXiv preprint (2021).
> >
> > \[5\] Chen, Jinsong, et al. "NAGphormer: A Tokenized Graph Transformer for Node Classification in Large Graphs." ICLR 2023\.
> >
> > \[6\] Wu, Qitian, et al. "Nodeformer: A scalable graph structure learning transformer for node classification." NeurIPS 2022\.
> >
> > \[7\] Chen, Jinsong, et al. "Rethinking Tokenized Graph Transformers for Node Classification." NeurIPS 2025\.
> >
> > \[8\] Xing, Yujie, et al. "Unifying and Enhancing Graph Transformers via a Hierarchical Mask Framework." NeurIPS 2025\.
> >
> > \[9\] Kong, Kezhi, et al. "GOAT: A global transformer on large-scale graphs." ICML 2023\.
> >
> > \[10\] Zhang, Zaixi, et al. "Hierarchical graph transformer with adaptive node sampling." NeurIPS 2022\.
> >
> > \[11\] Ma, Jiahong, Mingguo He, and Zhewei Wei. "Polyformer: Scalable node-wise filters via polynomial graph transformer." ACM SIGKDD 2024\.
> >
> > \[12\] Deng, Chenhui, Zichao Yue, and Zhiru Zhang. "Polynormer: Polynomial-Expressive Graph Transformer in Linear Time." ICLR 2024\.
> >
> > \[13\] Xing, Yujie, et al. "Less is More: on the Over-Globalizing Problem in Graph Transformers." ICML 2024\.

---

> > > ### Comment · Reviewer_3HAs · 2025-11-21
> > >
> > > Thank you for the authors’ response. I have read it carefully. The replies address most of my concerns, especially the additional experimental results, which further demonstrate the practical effectiveness of BTS. I hope these results can be included in the final version of the paper. However, the method's theoretical limitations and inherent complexity still somewhat affect its impact and applicability. Overall, I will raise my score to 6 and support the acceptance of this paper.

---

> > > > ### Author Response · Authors · 2025-11-21
> > > >
> > > > We thank the reviewer for their constructive feedback and for the updated score. We are glad that the revisions and additional results helped address the reviewer’s concerns, and we appreciate the time they dedicated to assessing our work.

---

### Official Review · Reviewer_Ay4a · 2025-10-27

**Soundness:** 3
**Presentation:** 3
**Contribution:** 2
**Rating:** 4
**Confidence:** 3

**Summary:**

This paper introduces Broaden the Spectrum (BTS), a data-driven method for selecting Laplacian eigenvectors as positional encodings (PEs) in Graph Transformers (GTs) for node classification. Existing GTs underperform on node classification due to their reliance on data-agnostic heuristics which ignore the graph-specific spectral distribution of class information. The main contributions of this paper are: (1) BTS Algorithm: A lightweight, task-aware method that selects eigenvectors most aligned with class labels using Energy Spectral Density (ESD). (2) Theoretical Justification: Shows that the optimal attention matrix for classification has a class-wise block structure, and that BTS-selected eigenvectors best approximate this structure. (3) Empirical Validation: Extensive results demonstrate significant performance gains across homophilic, heterophilic, and long-range benchmarks using standard GT architectures.

**Strengths:**

(1)	Novel & Principled Method: BTS is intuitive and theoretically grounded, bridging graph signal processing and transformer architectures, and is a simple yet effective method for improving GTs. Moreover, it is a plug-in module and can easily be applied into different backbones.

(2)	Solid Theoretical Analysis: It proves that optimal attention matrices for classification should have class-block structure.

(3)	Strong Empirical Results: Extensive experiment result, including large gains on challenging datasets; consistent improvements across multiple architectures and task types (homophily, heterophily, long-range).

**Weaknesses:**

(1)	Moderate novelty: This work can be seen as an incremental work on traditional positional encoding methods by adding ESD before selection of eigenvectors.

(2)	Large experimental searching space: According to Tab. 12, hyper-parameter searching space seems to be extremely large, which diminishing reproducibility of the results.

(3)	Clarification issue of motivations: Statement of data-agnostic methods seems ambiguous. (see Q.1, Q. 2)

(4)	Some issues and concerns of experimental parts: See Q. 3.

**Questions:**

(1)	Author states that current positional encoding methods are data-agnostic and proposed BTS is a data-adaptive method. However, calculation of eigenvector is still required in Algorithm 1, which I think is still related to graph data (structure). Does data-agnostic mean feature-free? This part should be discussed or clarified to avoid misunderstanding. Moreover, now that your method is data-adaptive, it is suggested to add comparison with other learnable positional encoding methods like LSPE [1].

(2)	For definition of ESD, it seems that it can be regarded as a preprocessing procedure which is static in essence. Then what is the difference between BTS and static positional encoding methods?

(3)	Please list computation cost and GPU consumption of BST and its comparison with LPE or other methods. It seems that ESD calculation is time-consuming, especially you claim that BST is “lightweight”. Moreover, what about its scalability on large-scale graphs?

[1] Dwivedi V P, Luu A T, Laurent T, et al. Graph neural networks with learnable structural and positional representations[J]. arXiv preprint arXiv:2110.07875, 2021.

---

> ### Author Response · Authors · 2025-11-21
> **Rebuttal (1/2)**
>
> We thank the reviewer for their time and comments. We are glad that the reviewer recognizes the main strengths of our paper: “Solid Theoretical Analysis,” “Principled Method,”  and “Strongical Empirical Results.” Most of the concerns raised relate to clarification of motivation, terminology, and experimental details. We address each point below and believe our responses resolve these concerns adequately.
>
> >**W1. Moderate novelty: This work can be seen as an incremental work on traditional positional encoding methods by adding ESD before selection of eigenvectors.**
>
> Yes, our final method is simple and builds on traditional LPEs. However, the novelty of our work extends well beyond the final algorithm. In particular:
>
> * **New theoretical framework:** We provide, to the best of our knowledge, the first principled characterization of what optimal attention matrices should look like for node classification. This, as you note in your review, is a major strength of our paper.
>
> * **Introduction of ESD:** The class-label energy spectral density (ESD) is a new conceptual tool that offers an interpretable view of how class information is distributed across the Laplacian spectrum and directly motivates our method.
>
> * **Explanation for GT underperformance:** We also are the first to show that the underperformance of LPE-based graph transformers on node classification stems primarily from previous *eigenvector selection heuristics*, and show that our principled selection strategy overcomes this bottleneck.
>
> These contributions collectively go beyond simply adding an extra step to traditional positional encodings.
>
> >**W2. Large experimental searching space: According to Tab. 12, hyper-parameter searching space seems to be extremely large, which diminishes reproducibility of the results.**
>
> We would like to clarify that our experiments remain fully reproducible and do not depend on an impractically large hyperparameter search, beyond what is standard in graph literature.
>
> * **No new hyperparameters.** BTS introduces *no* additional hyperparameters. These are the standard hyperparameters naturally present in the GT, Nagphormer, and GraphGPS models. Additionally, Table 12 aggregates the search spaces for GT, NAGphormer, and GraphGPS; not all hyperparameters apply to each model, as indicated by the headers below the horizontal lines.
>
> * **Reproducible tuning scripts.** Our supplementary code includes the exact hyperparameter tuning scripts used in our experiments, and reproduces our reported results.
>
> * **Efficient search.** We use Optuna’s Bayesian optimization, not grid search, so only a few hundred trials are sufficient to identify high-performing configurations. This process takes \~2-3 hours for each dataset on a 4 x NVidia L40S machine, which is comparable to other implementations. Further details for tuning are present in Appendix Section E of our manuscript.
>
> * **Fair comparison.** We use the similar search methods for BTS and for baselines, ensuring that comparisons remain fair and reproducible.
>
> Importantly, avoiding systematic tuning would actually reduce reproducibility. Relying on intuition or hand-picked hyperparameters would make results less rigorous, harder to replicate, and make models harder to adapt to new datasets.
>
> >**Q1a. Author states that current positional encoding methods are data-agnostic and proposed BTS is a data-adaptive method. However, calculation of eigenvector is still required in Algorithm 1, which I think is still related to graph data (structure). Does data-agnostic mean feature-free? This part should be discussed or clarified to avoid misunderstanding.**
>
> We appreciate the opportunity to clarify this point\! When we say that existing positional-encoding methods are *data-agnostic*, we mean that they use fixed, hand-crafted rules for selecting eigenvectors that do not depend on the label distribution (like top-k selection). This selection process is what we target by the phrase data-agnostic. The methods overall are dependent on the graph structure (as most structural and position encoding approaches are). BTS on the other hand, adapts the eigenvectors selection to the supervised task at hand.
>
> To illustrate this further, consider a fixed ring graph (cycle). Depending on how labels are assigned, this graph can represent either a *homophilic* task (adjacent nodes share the same label) or a *heterophilic* task (labels alternate around the cycle). Classical LPE methods would select the exact same subset of eigenvectors in both cases. BTS, however, would select more low-frequency eigenvectors for the homophilic case, and high-frequency eigenvectors for the alternating heterophilic case \- adapting to the task.

---

> > ### Author Response · Authors · 2025-11-21
> > **Rebuttal (2/2)**
> >
> > >**Q1b. Moreover, now that your method is data-adaptive, it is suggested to add comparison with other learnable positional encoding methods like LSPE \[1\].**
> >
> > We hope our response to Q1a clarifies what we mean by “adaptive.”
> >
> > Regarding comparisons with learnable positional encodings: Tables 1 and 2 in our paper already include models that incorporate learnable PE mechanisms. For example, both GT and GraphGPS allow positional embeddings to be updated throughout the network layers, which is precisely the core idea behind LSPE. In addition, GraphGPS includes a LapPE encoder, giving it even more flexibility to learn expressive positional representations.
> >
> > Thus, our baselines already cover models with learnable PEs, and BTS is evaluated fairly against these methods.
> >
> > >**Q2. For definition of ESD, it seems that it can be regarded as a preprocessing procedure which is static in essence. Then what is the difference between BTS and static positional encoding methods?**
> >
> > Thank you for raising this point. We believe there may be two interpretations of the term *“static positional encoding,”* so we clarify both.
> >
> > 1. **Static as “not updated during the forward pass.”** In this sense, no Laplacian PEs, including BTS, are generally used in a static manner: the eigenvector-based encodings are computed once and then passed through the model. These encodings are first processed by a linear layer or MLP, and the resulting embeddings are updated throughout the layers (similar to LSPE).
> >
> > 2. **Static as “task-agnostic or non-adaptive.”** This is the sense we refer to when contrasting BTS with prior work. Classical LPE methods use fixed heuristics (e.g., top-k) that select the *same* eigenvectors regardless of the supervised task. In contrast, BTS is adaptive: the subset of eigenvectors depends on the *class-label spectral energy* and thus changes with the supervised task. Two identical graphs with different label assignments would lead to different BTS-selected eigenvectors.
> >
> > We hope this resolves the confusion, and we would be happy to clarify further if needed.
> >
> > >**Q3. Please list computation cost and GPU consumption of BTS and its comparison with LPE or other methods. It seems that ESD calculation is time-consuming, especially you claim that BTS is “lightweight”. Moreover, what about its scalability on large-scale graphs?**
> >
> > Given the Laplacian eigenvectors, the computation of ESD and the subsequent thresholding step are extremely lightweight. The most expensive operation is a *single matrix multiplication* (line 2 in Algorithm 1: $\\tilde{Y}\_\\text{train} \= V\_\\text{train}^\\top Y\_\\text{train}$), which runs on the GPU and is **negligible compared to the overall training time.** For instance, our training of GT-BTS on the Coauthor-Physics dataset takes \~20 seconds, while the BTS step contributes to only 0.52 seconds. Thus, the additional overhead compared to standard LPE is negligible.
> >
> > As with all eigenvector-based methods, the primary computational bottleneck remains the eigenvector computation itself, and not our selection procedure. For large graphs, we follow previous practice \[1\] and do not compute the full spectrum. Instead, for graphs with more than 8192 nodes, we compute only the first and last 4096 eigenvectors, as mentioned in the footnote on Page 6 of our manuscript.
> >
> > ---
> > **References**
> >
> > \[1\] Bo, Deyu, et al. "Specformer: Spectral Graph Neural Networks Meet Transformers." The Eleventh International Conference on Learning Representations (2023)

---

> > > ### Author Response · Authors · 2025-11-21
> > > **Additional Updates**
> > >
> > > In addition to our point-by-point response above, we are excited to share several new results and invite you to read our General Rebuttal Response for an overview. To summarize,
> > >
> > > 1. We have added results from four more baselines: **PolyFormer** \[2\], **Specformer** \[1\], **Polynormer** \[3\], and **CoBFormer** \[4\] (as suggested by reviewers wJCD and 3HAs), and compared all models using average rank. These new results have been added to **Tables 1 and 2 (page 7\) and Figure 3 (page 8\) of the updated manuscript** (highlighted in blue). Overall, the BTS variants of GT, NAGphormer, and GraphGPS remain highly competitive with these recent state-of-the-art models, with GraphGPS-BTS achieving **best rank overall across all datasets**. This supports our central claim that LPE-based graph transformers were primarily held back by the choice of eigenvectors.
> > >
> > > 2. We ran an additional experiment in which we artificially restricted the number of training labels in GT and GT-BTS to test the impact of label scarcity. We have incorporated this experiment into **Appendix B (page 14\) and Figure 5 (page 15\) of the revised manuscript.** We find that GT-BTS consistently outperforms the baseline GT across all datasets and label budgets. This shows that **even in label-scarce regimes, BTS-selected eigenvectors remain more effective** than the classical top-k selection.
> > >
> > > We hope these additional results will better contextualize the effectiveness of BTS against recent state-of-the-art methods and under varying label conditions.
> > >
> > > ---
> > >
> > > **References**
> > >
> > > [2] Ma, Jiahong, Mingguo He, and Zhewei Wei. "Polyformer: Scalable node-wise filters via polynomial graph transformer." Proceedings of the 30th ACM SIGKDD Conference on Knowledge Discovery and Data Mining. 2024.
> > >
> > > [3] Deng, Chenhui, Zichao Yue, and Zhiru Zhang. "Polynormer: Polynomial-Expressive Graph Transformer in Linear Time." The Twelfth International Conference on Learning Representations (2024).
> > >
> > > [4] Xing, Yujie, et al. "Less is More: on the Over-Globalizing Problem in Graph Transformers." Forty-first International Conference on Machine Learning (2024).

---

### Official Review · Reviewer_m6Ny · 2025-11-01

**Soundness:** 2
**Presentation:** 2
**Contribution:** 2
**Rating:** 4
**Confidence:** 4

**Summary:**

This paper examines why graph transformers often underperform on node classification tasks, identifying the choice of eigenvectors for Laplacian positional encodings as a key factor. To address this, the authors introduce Broaden the Spectrum (BTS), a data-driven approach that selects eigenvectors based on their alignment with class label energy. Theoretical analysis explains how BTS promotes attention matrices with class-aligned block structures, and extensive experiments on homophilic, heterophilic, and long-range benchmarks demonstrate that BTS significantly outperforms common eigenvector selection heuristics.

**Strengths:**

1. Figure 1 offers a clear illustration of class-label energy distributions, highlighting the importance of adaptive spectrum selection.

2. The theoretical analysis helps to understand the method’s underlying principles.

3. Experiments across diverse graph types, particularly heterophilic and long-range datasets, demonstrate the effectiveness of the proposed approach.

**Weaknesses:**

1. Additional related works [1-2] on adaptive or alternative positional encodings in graph transformers should be discussed to provide a more comprehensive context.

2. The proposed use of label-aligned spectral energy for positional encoding selection relies heavily on labeled data, which may not always be readily available.

3. While the paper focuses on node classification, it would be valuable to explore whether the proposed approach can generalize to graph-level classification tasks.

[1] Park, Wonpyo, et al. "Grpe: Relative positional encoding for graph transformer." arXiv preprint arXiv:2201.12787 (2022).

[2] Li, Chenyang, et al. "DAM-GT: Dual Positional Encoding-Based Attention Masking Graph Transformer for Node Classification." arXiv preprint arXiv:2505.17660 (2025).

**Questions:**

How does the proposed method perform under label-scarce settings?

---

> ### Author Response · Authors · 2025-11-21
> **Rebuttal (1/2)**
>
> We thank the reviewer for their time and feedback. We appreciate that the reviewer recognized the “importance of adaptive spectrum selection,” and that our “theoretical analysis helps to understand the method’s underlying principles.” We provide a point-by-point response to your comments and concerns below.
>
> >**W1. Additional related works \[1-2\] on adaptive or alternative positional encodings in graph transformers should be discussed to provide a more comprehensive context.**
>
> Thank you for pointing out this related work\! We have updated the Related Works section of our paper, and added these two works (lines 475-484 on page 9 in the updated PDF).
>
> >**W2. The proposed use of label-aligned spectral energy for positional encoding selection relies heavily on labeled data, which may not always be readily available.**
>
> Thank you for bringing up this concern. Our dependence on labels for eigenvector selection indeed raises the question of whether BTS continues to outperform classical top-k selection when labels are scarce. This sentiment is also shared by reviewer wJCD. To address this, we ran an additional experiment in which we artificially restrict the number of training labels. We measure performances on this label-scarce setting on four datasets and report comparisons between GT and GT-BTS below.
>
> As expected, the test accuracy of both models decreases as the number of training labels is reduced. However, GT-BTS still consistently outperforms the baseline GT across all datasets and label budgets. This shows that even in label-scarce regimes, BTS-selected eigenvectors remain more effective than the classical top-k selection.
> **We have incorporated this experiment into Appendix B (page 14) and Figure 5 (page 15) of the revised manuscrip**t.
>
> **Table** **R2A**: Comparison of GT and GT-BTS in the label-scarce setting. GT-BTS consistently outperforms GT despite decreasing label-budget.
>
> |  | Chameleon | Chameleon | Squirrel | Squirrel | WikiCS | WikiCS | Computers | Computers |
> | ----- | ----- | ----- | ----- | ----- | ----- | ----- | ----- | ----- |
> | **Training Label Percentage** | **GT** | **GT-BTS** | **GT** | **GT-BTS** | **GT** | **GT-BTS** | **GT** | **GT-BTS** |
> | 100% | 50.48 ± 2.08 | **73.09 ± 1.00** | 34.70 ± 1.77 | **65.06 ± 1.93** | 72.91 ± 0.59 | **78.94 ± 0.26** | 85.65 ± 0.59 | **91.46 ± 0.51** |
> | 75% | 49.25 ± 2.58 | **69.52 ± 1.78** | 34.78 ± 1.72 | **60.45 ± 2.41** | 71.37 ± 0.86 | **77.45 ± 0.68** | 85.52 ± 0.54 | **91.31 ± 0.45** |
> | 50% | 47.76 ± 3.34 | **65.64 ± 2.66** | 33.78 ± 1.42 | **55.03 ± 1.61** | 68.85 ± 0.95 | **76.05 ± 0.53** | 83.96 ± 0.66 | **90.75 ± 0.57** |
> | 25% | 44.63 ± 2.84 | **60.92 ± 1.72** | 33.23 ± 1.23 | **46.95 ± 0.99** | 64.81 ± 1.54 | **71.54 ± 1.44** | 82.41 ± 0.73 | **89.15 ± 0.58** |
>
> >**W3. While the paper focuses on node classification, it would be valuable to explore whether the proposed approach can generalize to graph-level classification tasks.**
>
> We would like to clarify the motivation and scope of our work. We focus specifically on **node classification**, because this is the setting where graph transformers have historically struggled to match the performance of MPNNs and spectral GNNs \[1, 2\]. Our goal is to demonstrate that graph transformers, particularly LPE-based ones, have been held back primarily by *how* eigenvectors were selected, and that with a principled selection strategy, they can be highly competitive in the node-classification regime.
>
> Other tasks pose different challenges for injecting positional or structural information, and have been addressed extensively in prior work \[3, 4\]. Our contribution is therefore focused and complementary: we provide a theoretical framework and a practical solution tailored to node classification, the regime where graph transformers have been most disadvantaged. There is also strong precedent for this focus, as several prior works have studied only node classification for the same reason \[5, 6, 7, 8, 9, 10\]
>
> While it is intriguing to consider how one might select LPEs for graph-level classification, we leave this for future work, and have already mentioned this in the Conclusion section of our paper.
>
> >**Q1. How does the proposed method perform under label-scarce settings?**
>
> Please see our response to W2 above.

---

> > ### Author Response · Authors · 2025-11-21
> > **Rebuttal (2/2)**
> >
> > **References**
> >
> > \[1\] Bo, Deyu, et al. "Specformer: Spectral Graph Neural Networks Meet Transformers." ICLR 2023\.
> >
> > \[2\] Luo, Yuankai, Lei Shi, and Xiao-Ming Wu. "Classic GNNs are Strong Baselines: Reassessing GNNs for Node Classification." NeurIPS 2024\.
> >
> > \[3\] Shomer, Harry, et al. "Lpformer: An adaptive graph transformer for link prediction." ACM SIGKDD 2024\.
> >
> > \[4\] Mialon, Grégoire, et al. "Graphit: Encoding graph structure in transformers." arXiv preprint (2021).
> >
> > \[5\] Chen, Jinsong, et al. "NAGphormer: A Tokenized Graph Transformer for Node Classification in Large Graphs." ICLR 2023\.
> >
> > \[6\] Wu, Qitian, et al. "Nodeformer: A scalable graph structure learning transformer for node classification." NeurIPS 2022\.
> >
> > \[7\] Chen, Jinsong, et al. "Rethinking Tokenized Graph Transformers for Node Classification." NeurIPS 2025\.
> >
> > \[8\] Xing, Yujie, et al. "Unifying and Enhancing Graph Transformers via a Hierarchical Mask Framework." NeurIPS 2025\.
> >
> > \[9\] Kong, Kezhi, et al. "GOAT: A global transformer on large-scale graphs." ICML 2023\.
> >
> > \[10\] Zhang, Zaixi, et al. "Hierarchical graph transformer with adaptive node sampling." NeurIPS 2022\.

---

> > > ### Author Response · Authors · 2025-11-21
> > > **Additional Updates**
> > >
> > > In addition to the label scarcity analysis in our point-by-point response above, we are excited to share additional new results and invite you to read our General Rebuttal Response for an overview.
> > >
> > > To summarize, we have added results from four more baselines: **PolyFormer** [11], **Specformer** [1], **Polynormer** [12], and **CoBFormer** [13] (as suggested by reviewers wJCD and 3HAs), and compared all models using average rank. These new results have been added to **Tables 1 and 2 (page 7) and Figure 3 (page 8) of the updated manuscript** (highlighted in blue).
> > >
> > > Overall, the BTS variants of GT, NAGphormer, and GraphGPS remain highly competitive with these recent state-of-the-art models, with GraphGPS-BTS achieving **best rank overall across all datasets**. This supports our central claim that LPE-based graph transformers were primarily held back by the choice of eigenvectors.
> > >
> > > ---
> > >
> > > **References**
> > >
> > > [11] Ma, Jiahong, Mingguo He, and Zhewei Wei. "Polyformer: Scalable node-wise filters via polynomial graph transformer." Proceedings of the 30th ACM SIGKDD Conference on Knowledge Discovery and Data Mining. 2024.
> > >
> > > [12] Deng, Chenhui, Zichao Yue, and Zhiru Zhang. "Polynormer: Polynomial-Expressive Graph Transformer in Linear Time." The Twelfth International Conference on Learning Representations (2024).
> > >
> > > [13] Xing, Yujie, et al. "Less is More: on the Over-Globalizing Problem in Graph Transformers." Forty-first International Conference on Machine Learning (2024).

---

### Official Review · Reviewer_wJCD · 2025-11-01

**Soundness:** 2
**Presentation:** 2
**Contribution:** 3
**Rating:** 4
**Confidence:** 5

**Summary:**

This paper study how the selection of Laplacian eigenvectors as positional encodings influences graph transformer performance in node classification. The authors propose Broaden the Spectrum (BTS) to selects eigenvectors according to their class-label energy spectral density (ESD). They show that the optimal attention matrix has a class-wise block structure and that high-label-energy eigenvectors best approximate it. Experiments demonstrate consistent large performance gains on heterophilic and long-range benchmarks, across several graph transformer architectures.

**Strengths:**

1. Introducing adaptive frequency selection meaningfully advances positional encoding in graph transformers.

2. The model is simple, effective, and supported by theoretical analysis.

**Weaknesses:**

1. While early graph transformers relied heavily on Laplacian eigenvectors to incorporate graph topology, recent work has demonstrated that structural biases can also be introduced through GNNs [1,2,3] and attention masks[4]. Thus, the statements "graph
transformers require explicit positional encodings to inject structural information" in Abstract and "transformers rely on positional encodings (PEs) to inject structural information" in Introduction may introduce some misleading understanding.


2. While the proposed BTS is theoretically grounded in the concept of class-label energy spectral density (ESD), this dependence constrains its applicability to supervised settings. Therefore, the impact of label quantity on performance needs to be studied.

3. The explanation of the “class-wise block structure” is somewhat ambiguous. Intuitively, the optimal attention matrix is purely block-diagonal form (where the diagonal blocks are non-zero and the off-diagonal blocks are zero). In Figure 2, the empirical pattern shows substantial inter-class attention, which seems inconsistent with the intuitive clustering.

4. The paper omits comparison and discussion with several state-of-the-art transformer-based graph models, including Polynormer [1], CoBFormer [2], DualFormer [3], and Gradformer [4], which represent the latest advances in structural bias integration. This omission substantially weakens the empirical credibility of the paper’s claimed contributions.

[1] Polynomial-Expressive Graph Transformer in Linear Time, in ICLR 24.

[2] Less is More: on the Over-Globalizing Problem in Graph Transformers, In ICML 24.

[3] DUALFormer: Dual Graph Transformer, in ICLR 25.

[4] Gradformer: Graph Transformer with Exponential Decay, in IJCAI 24.

**Questions:**

See Weaknesses.

---

> ### Author Response · Authors · 2025-11-21
> **Rebuttal (1/3)**
>
> We sincerely thank the reviewer for their thoughtful and constructive feedback. We are glad the reviewer recognizes that our paper “meaningfully advances positional encoding in graph transformers,” and that our method “is simple, effective, and supported by theoretical analysis.” We present a point-by-point response to the reviewer’s concerns and comments below.
>
> > **W1. While early graph transformers relied heavily on Laplacian eigenvectors to incorporate graph topology, recent work has demonstrated that structural biases can also be introduced through GNNs \[1,2,3\] and attention masks\[4\]. Thus, the statements \[..\] may introduce some misleading understanding.**
>
> We agree that Laplacian positional encodings (LPEs) are not the only mechanism for injecting structural information into graph transformers. As you mention, recent works introduce structure through attention biases or GNN-based modules. LPEs were among the earliest approaches, but structural attention biases later became more prominent because they achieved stronger empirical performance on several benchmarks.
>
> We argue this performance gap stemmed not from inherent limitations of LPEs, but from the simplistic heuristics used in early implementations. Once this component is replaced with a principled selection strategy (as we propose with BTS) we find that LPE-based graph transformers can be substantially more effective than previously shown.
>
> LPEs also offer a practical advantage as they are compatible with high-performance attention kernels (e.g., [FlashAttention](https://github.com/Dao-AILab/flash-attention), [xFormers' MemoryEfficientAttention](https://github.com/facebookresearch/xformers)), whereas attention-bias methods typically incur significant runtime and memory overhead simply due to the nature of these kernels. Strengthening LPE-based approaches therefore has strong practical implications as well.
>
> To avoid giving the impression that PEs are the *only* way to inject structure, we have revised the Abstract (lines 014–017), Introduction (lines 040–050), and Related Works (lines 475-484 on page 9) to more accurately reflect this broader landscape.
>
> > **W2. The impact of label quantity on performance needs to be studied.**
>
> Thank you for bringing up this concern. Our dependence on labels for eigenvector selection indeed raises the question of whether BTS continues to outperform classical top-k selection when labels are scarce. To address this, we ran an additional experiment in which we artificially restrict the number of training labels. We measure performances on this label-scarce setting on four datasets and report comparisons between GT and GT-BTS below.
>
> As expected, the test accuracy of both models decreases as the number of training labels is reduced. However, GT-BTS still consistently outperforms the baseline GT across all datasets and label budgets. This shows that even in label-scarce regimes, BTS-selected eigenvectors remain more effective than the classical top-k selection.
> **We have incorporated this experiment into Appendix B (page 14) and Figure 5 (page 15) of the revised manuscrip**t.
>
> **Table** **R1A**: Comparison of GT and GT-BTS in the label-scarce setting. GT-BTS consistently outperforms GT despite decreasing label-budget.
>
> |  | Chameleon | Chameleon | Squirrel | Squirrel | WikiCS | WikiCS | Computers | Computers |
> | :---: | :---: | :---: | :---: | :---: | :---: | :---: | :---: | :---: |
> | **Training Label Percentage** | **GT** | **GT-BTS** | **GT** | **GT-BTS** | **GT** | **GT-BTS** | **GT** | **GT-BTS** |
> | 100% | 50.48 ± 2.08 | **73.09 ± 1.00** | 34.70 ± 1.77 | **65.06 ± 1.93** | 72.91 ± 0.59 | **78.94 ± 0.26** | 85.65 ± 0.59 | **91.46 ± 0.51** |
> | 75% | 49.25 ± 2.58 | **69.52 ± 1.78** | 34.78 ± 1.72 | **60.45 ± 2.41** | 71.37 ± 0.86 | **77.45 ± 0.68** | 85.52 ± 0.54 | **91.31 ± 0.45** |
> | 50% | 47.76 ± 3.34 | **65.64 ± 2.66** | 33.78 ± 1.42 | **55.03 ± 1.61** | 68.85 ± 0.95 | **76.05 ± 0.53** | 83.96 ± 0.66 | **90.75 ± 0.57** |
> | 25% | 44.63 ± 2.84 | **60.92 ± 1.72** | 33.23 ± 1.23 | **46.95 ± 0.99** | 64.81 ± 1.54 | **71.54 ± 1.44** | 82.41 ± 0.73 | **89.15 ± 0.58** |

---

> ### Author Response · Authors · 2025-11-21
> **Rebuttal (2/3)**
>
> >**W3. The explanation of the “class-wise block structure” is somewhat ambiguous. Intuitively, the optimal attention matrix is purely block-diagonal form (where the diagonal blocks are non-zero and the off-diagonal blocks are zero). In Figure 2, the empirical pattern shows substantial inter-class attention, which seems inconsistent with the intuitive clustering.**
>
> We appreciate the opportunity to clarify this point. Intuitively, an attention matrix that is optimal for classification must satisfy *two* complementary objectives:
>
> 1. **Within-class clustering:** node embeddings of the same class should be tightly clustered together (increasing the numerator of the cross-entropy softmax).
> 2. **Inter-class separation:** the separation between these class-clusters should be maximized (decreasing denominator of the cross-entropy softmax).
>
> A purely block-diagonal attention matrix satisfies only the first objective: it encourages within-class clustering but provides no mechanism for increasing inter-class separation. However, achieving the second objective requires off-diagonal blocks to be non-zero as well, where interactions between classes help position clusters farther from each other in the embedding space. And so, in most cases, the optimal attention matrix will not be a pure block-diagonal matrix.
>
> We updated the paper with this intuition (lines 252-258 on page 5\). We would be happy to clarify any further\!

---

> ### Author Response · Authors · 2025-11-21
> **Rebuttal (3/3)**
>
> >**W4. The paper omits comparison and discussion with several state-of-the-art transformer-based graph models, including Polynormer \[1\], CoBFormer \[2\], DualFormer \[3\], and Gradformer \[4\], which represent the latest advances in structural bias integration. This omission substantially weakens the empirical credibility of the paper’s claimed contributions.**
>
> Thank you for pointing this out\! We have now added results from four additional baselines: **Polynormer** \[1\], **CoBFormer** \[2\], **SpecFormer** \[5\], and **PolyFormer** \[6\] (the latter two requested by reviewer 3HAs). We were unable to include DualFormer because no code has been released; and GradFormer has only been evaluated on graph classification. Because published results were not available for all datasets, we reproduced these methods ourselves, ensuring a fair comparison under the same data splits and a similar hyperparameter-tuning setup. These new results have been added to **Tables 1 and 2 (page 7\) and Figure 3 (page 8\) of the updated manuscript** (highlighted in blue), and we present a focused set of results below.
>
> Based on **average rank** (lower is better), we observe the following trends:
>
> - **Heterophilic datasets:** `GraphGPS-BTS < NAGphormer-BTS < Polynormer < GT-BTS < GraphGPS < … `
> - **Homophilic datasets:** `PolyFormer < GraphGPS-BTS < NAGphormer-BTS < SpecFormer < GT-BTS < …`
> - **Overall:** `GraphGPS-BTS < PolyFormer < NAGphormer-BTS < Polynormer < GT-BTS < …`
>
> GraphGPS goes from being ranked 6th overall to 1st, when equipped with BTS. Notably, GT, the simplest transformer in our study, improves dramatically, going from being ranked 12th overall to 5th. Overall, the BTS variants of GT, NAGphormer, and GraphGPS remain highly competitive with these recent state-of-the-art models. This further supports our central claim: LPE-based graph transformers were primarily held back by the choice of eigenvectors, and with a principled selection strategy, even simple architectures can achieve strong node-classification performance.
>
> **Table R1B:** Performance on heterophilic datasets
>
> | Model | Chameleon | Squirrel | Chameleon (filtered) | Squirrel (filtered) | Tolokers | Ratings | Average Rank |
> | :---: | :---: | :---: | :---: | :---: | :---: | :---: | :---: |
> | GT-BTS | 73.09 ± 1.00 | 65.06 ± 1.93 | 45.51 ± 4.69 | 40.32 ± 1.63 | 84.45 ± 0.66 | 50.37 ± 0.48 | 5.17 |
> | NAGphormer-BTS | 73.90 ± 1.68 | 65.04 ± 1.69 | 49.01 ± 4.04 | 43.24 ± 2.83 | 85.47 ± 0.72 | 49.65 ± 0.65 | 4.00 |
> | GraphGPS-BTS | 73.16 ± 1.70 | 65.87 ± 1.30 | 44.26 ± 3.99 | 44.74 ± 2.10 | 86.31 ± 0.63 | 51.33 ± 0.58 | 3.17 |
> | Polyformer | 63.75 ± 1.52 | 43.19 ± 2.18 | 45.49 ± 3.35 | 42.72 ± 2.25 | 85.11 ± 0.84 | 50.02 ± 0.54 | 5.83 |
> | Polynormer | 74.34 ± 1.98 | 66.91 ± 2.31 | 43.53 ± 3.20 | 42.71 ± 2.23 | 84.52 ± 0.29 | 52.72 ± 0.54 | 4.33 |
> | CoBformer | 53.82 ± 1.51 | 38.36 ± 1.81 | 45.48 ± 1.52 | 40.14 ± 1.01 | 81.85 ± 0.24 | 50.83 ± 0.59 | 8.50 |
> | Specformer | 73.49 ± 1.87 | 63.65 ± 1.88 | 40.57 ± 3.63 | 39.65 ± 1.62 | 80.86 ± 0.85 | 51.54 ± 0.28 | 7.83 |
>
> **Table R1C:** Performance on homophilic datasets
>
> | Model | Physics | CS | Photo | Computers | WikiCS | ogbn-arXiv | Average Rank |
> | :---: | :---: | :---: | :---: | :---: | :---: | :---: | :---: |
> | GT-BTS | 96.90 ± 0.18 | 95.44 ± 0.33 | 95.95 ± 0.48 | 91.46 ± 0.51 | 78.94 ± 0.26 | 70.30 ± 0.12 | 6.83 |
> | NAGphormer-BTS | 97.05 ± 0.18 | 95.42 ± 0.39 | 95.90 ± 0.37 | 91.85 ± 0.44 | 79.42 ± 0.55 | 71.29 ± 0.13 | 4.83 |
> | GraphGPS-BTS | 97.21 ± 0.14 | 95.72 ± 0.37 | 95.87 ± 0.42 | 91.87 ± 0.45 | 79.47 ± 0.48 | 70.92 ± 0.33 | 3.67 |
> | Polyformer | 97.06 ± 0.24 | 95.57 ± 0.31 | 95.98 ± 0.51 | 92.22 ± 0.45 | 79.69 ± 0.45 | 71.73 ± 0.28 | 3.00 |
> | Polynormer | 96.61 ± 0.23 | 95.28 ± 0.37 | 95.58 ± 0.61 | 91.74 ± 0.64 | 79.16 ± 0.68 | 71.23 ± 0.27 | 7.00 |
> | CoBformer | 96.49 ± 0.08 | 94.99 ± 0.14 | 94.12 ± 0.33 | 90.87 ± 0.27 | 80.26 ± 0.21 | 72.69 ± 0.12 | 8.00 |
> | Specformer | 97.04 ± 0.20 | 95.84 ± 0.40 | 92.18 ± 0.62 | 92.87 ± 0.30 | 80.23 ± 0.64 | 70.20 ± 0.15 | 5.33 |
>
> ---
> **References**
>
> \[1\]  Deng, Chenhui, Zichao Yue, and Zhiru Zhang. "Polynormer: Polynomial-Expressive Graph Transformer in Linear Time." The Twelfth International Conference on Learning Representations (2024)
>
> \[2\] Xing, Yujie, et al. "Less is More: on the Over-Globalizing Problem in Graph Transformers." Forty-first International Conference on Machine Learning (2024)
>
> \[5\]  Bo, Deyu, et al. "Specformer: Spectral Graph Neural Networks Meet Transformers." The Eleventh International Conference on Learning Representations (2023)
>
> \[6\] Ma, Jiahong, Mingguo He, and Zhewei Wei. "Polyformer: Scalable node-wise filters via polynomial graph transformer." Proceedings of the 30th ACM SIGKDD Conference on Knowledge Discovery and Data Mining. 2024\.

---

> > ### Comment · Reviewer_wJCD · 2025-11-24
> >
> > Thank you for the detailed response. After carefully reviewing the revised manuscript, I am pleased to see that W1 and W4 have been largely addressed. I would just like to remind the authors that the code repository for Dualformer appears to be: https://github.com/JiamingZhuo/DUALFormer.
> >
> > I still find the responses to W2 and W3 insufficiently clear.
> >
> > For W2, I believe that a low-label setting is more representative of realistic scenarios. It is therefore critical to verify whether the proposed model remains effective when only a few labels per class are available, as this is key to assessing its practical applicability and overall soundness. I suggest that the authors additionally report results like using 3, 5, and 10 labeled nodes per class.
> >
> > For W3, I see a mismatch between the authors' interpretation and the experimental evidence. I agree that within-class clustering and Inter-class separation are important objectives, but the proposed model does not seem to achieve both. In particular, in Figure 4, many diagonal blocks are predominantly strong negative (for example, in Chameleon–GT_{GTS}), which in no way achieves intra-class clustering and is even more detrimental than GT (with weak negative).

---

> > > ### Author Response · Authors · 2025-11-26
> > >
> > > Thanks for your quick response! We are glad we were able to address concerns regarding W1 and W4. Below, we provide further results and clarifications for W2 and W3.
> > >
> > > > **For W2, I believe that a low-label setting is more representative of realistic scenarios. It is therefore critical to verify whether the proposed model remains effective when only a few labels per class are available, as this is key to assessing its practical applicability and overall soundness. I suggest that the authors additionally report results like using 3, 5, and 10 labeled nodes per class.**
> > >
> > > As requested, we have run GT and GT-BTS on four datasets with 3, 5, and 10 labeled nodes per class. The results are reported below. As expected, the BTS variant performs better or stays comparable with GT (within the margin of error).
> > >
> > > **Table R1D:** Few-shot evaluation of GT and GT-BTS.
> > > | Num labels per class | Chameleon | Chameleon | Squirrel | Squirrel | WikiCS | WikiCS | Computers | WikiCS |
> > > | ----- | ----- | ----- | ----- | ----- | ----- | ----- | ----- | ----- |
> > > |  | GT | GT-BTS | GT | GT-BTS | GT | GT-BTS | GT | GT-BTS |
> > > | 3 | **37.87 ± 2.92** | 37.33 ± 3.07 | 31.88 ± 1.75 | **32.37 ± 1.59** | 47.53 ± 3.71 | **62.06 ± 3.00** | 38.63 ± 1.95 | **67.70 ± 4.41** |
> > > | 5 | 39.17 ± 1.90 | **40.81 ± 3.75** | 31.73 ± 1.31 | **31.91 ± 1.51** | 53.19 ± 4.20 | **65.53 ± 3.15** | 45.80 ± 3.74 | **73.32 ± 2.26** |
> > > | 10 | 39.45 ± 2.37 | **44.10 ± 3.17** | 31.60 ± 2.61 | **32.35 ± 1.21** | 61.27 ± 2.95 | **68.89 ± 2.09** | 55.75 ± 4.19 | **76.73 ± 2.34** |
> > >
> > > We would, however, like to note that this *very* low-label few-shot setting is not standard for supervised models. Most prior work on supervised node classification (including those cited by the reviewer) evaluates under the established benchmark splits we have already used in our main Results section. Few-shot evaluations of this kind are typically used to assess *self-supervised* or *foundation-model* approaches, which is not the focus of our method.
> > >
> > > > **For W3, I see a mismatch between the authors' interpretation and the experimental evidence. I agree that within-class clustering and Inter-class separation are important objectives, but the proposed model does not seem to achieve both. In particular, in Figure 4, many diagonal blocks are predominantly strong negative (for example, in Chameleon–GT\_{BTS}), which in no way achieves intra-class clustering and is even more detrimental than GT (with weak negative).**
> > >
> > > Thank you for pointing this out, we agree that negative values on the block diagonal, coupled with row-wise softmax that the attention operation performs, would mean that *that particular* GT-BTS’s attention matrix does not lead to much intra-class clustering.
> > >
> > > However, the situation is more complicated in real transformers which have multiple attention layers (each with multiple heads) as well as feedforward networks. With this complexity, different operations (intra-class clustering and inter-class separation *across different classes*) can be distributed across layers and heads; they don’t need to happen in one single attention operation. In addition, the transformer’s feedforward networks can also collapse or separate class-wise clusters independent of the attention heads. We have added **Appendix E in the updated manuscript (page 20\)** with figures showing that each attention head across each layer has a different block structure, depicting the distributed nature of the computation.
> > >
> > > The intuition we provided is mainly applicable to our *theoretical* model (single attention layer and head, no softmax, no feedforward network). Translating that intuition to deep transformers is indeed imperfect, as you pointed out. That said, we *do* observe both the emergence of block structure and the performance improvements happening together.
> > >
> > > We would also like to note that in the Chameleon-GT attention matrix in Figure 4, the values are almost all negative; they appear to be positive because of the color scale. Since interpreting the attention scores without softmax can be non-trivial, we have replaced Fig 4 to include attention matrices with softmax normalization applied. For reference, we have also attached the previous version of the plot [here](https://ik.imagekit.io/0k5zsui0s/attention_matrices.png).

---

### Author Response · Authors · 2025-11-21
**General Rebuttal Response (1/2)**

We sincerely thank all reviewers for their thoughtful and constructive feedback. We are pleased that the core motivations, theoretical insights, and empirical strengths of our work were broadly recognized across all reviews.

* **Clear Motivation & Core Insight:** Reviewers noted that “*introducing adaptive frequency selection meaningfully advances positional encoding in graph transformers.*” (wJCD and Ay4a), and that “*Figure 1 offers a clear illustration of class-label energy distributions, highlighting the importance of adaptive spectrum selection.*” (m6Ny)
* **Theoretical Contributions:** Reviewers noted that our analysis “*helps to understand the method’s underlying principles*” (m6Ny), it “*proves that optimal attention matrices for classification should have class-block structure*” (Ay4a), and that our theory “*elucidates its effectiveness in the context of node classification tasks*” (3HAs).
* **Simplicity, Practicality & Adaptability of BTS:** Reviewers highlighted that BTS is “*simple, effective*” (wJCD), “*intuitive and theoretically grounded \[…\] and can easily be applied into different backbones*” (Ay4a), and “*simple, intuitive, and easily adaptable to a wide range of graph transformer models*” (3HAs).
* **Strong Empirical Results:** Reviewers emphasized our “*strong gains on heterophilic and long-range datasets*” (m6Ny), praised the “*extensive experiment result, including large gains on challenging datasets*” (Ay4a), and commended the “*extensive ablation studies that validate the efficacy of the proposed BTS method*” (3HAs).

\
We take this opportunity to address some concerns shared by multiple reviewers:

**1. Performance study under label-scarce setting:** To address reviewer wJCD and m6Ny’s concerns about the performance of our method in label scarce settings, we conducted an additional experiment in which we artificially restrict the number of training labels. **This analysis has been added to Appendix B in our updated manuscript**. (Results also in table R1A in our response to reviewer wJCD)

As expected, the test accuracy of both models decreases as the number of training labels is reduced. However, GT-BTS still consistently outperforms the baseline GT across all label budgets, showing that even in label-scarce regimes, selecting eigenvectors using BTS remains more effective than the classical top-k selection.

\
**2. Extended Baseline Comparisons:** As requested by reviewers wJCD and 3HAs, we now include comparisons with four recent state-of-the-art models: PolyFormer, CoBFormer, and Polynormer (graph transformers), and SpecFormer (a spectral GNN). Because the literature does not report results on all of our datasets, we reran these baselines under our standardized training pipeline and hyperparameter-tuning setup to ensure a fair and consistent comparison. These **new results have been added to Tables 1 and 2 (page 7\) and Figure 3 (page 8\) of the updated manuscript** (highlighted in blue).

Based on average rank (lower is better), we observe the following trends:

- **Heterophilic datasets:** `GraphGPS-BTS < NAGphormer-BTS < Polynormer < GT-BTS < GraphGPS < … `
- **Homophilic datasets:** `PolyFormer < GraphGPS-BTS < NAGphormer-BTS < SpecFormer < GT-BTS < …`
- **Overall:** `GraphGPS-BTS < PolyFormer < NAGphormer-BTS < Polynormer < GT-BTS < …`

GraphGPS goes from being ranked 6th overall to 1st, when equipped with BTS. Notably, GT, the simplest transformer in our study, improves dramatically, going from being ranked 12th overall to 5th. Overall, the BTS variants of GT, NAGphormer, and GraphGPS not only outperform their respective base models but also remain highly competitive with recent state-of-the-art transformer and spectral approaches.

\
**3. On graph classification and other tasks:** Several reviewers note that our analysis does not directly extend to tasks such as graph classification or link prediction. We would like to clarify the motivation and scope of our work. We focus specifically on **node classification**, because this is the setting where graph transformers have historically struggled to match the performance of MPNNs and spectral GNNs \[1, 2\].

Our goal is to demonstrate that graph transformers, particularly LPE-based ones, have been held back primarily by *how* eigenvectors were selected, and that with a principled selection strategy, they can be highly competitive in the node-classification regime.

Other tasks pose different challenges for injecting positional or structural information, and have been addressed extensively in prior work \[3, 4\]. Our contribution is therefore focused and complementary: we provide a theoretical framework and a practical solution tailored to node classification, the regime where graph transformers have been most disadvantaged. There is also strong precedent for this focus, as several prior works have studied only node classification for similar reasons \[5, 6, 7, 8, 9, 10\].

---

> ### Author Response · Authors · 2025-11-21
> **General Rebuttal Response (2/2)**
>
> We have responded point-by-point to each reviewer, and have incorporated clarifications and new experiments into the manuscript. We hope our individual responses adequately address the various concerns brought up by the reviewers\!
>
> ---
> **References**
>
> \[1\] Bo, Deyu, et al. "Specformer: Spectral Graph Neural Networks Meet Transformers." ICLR 2023\.
>
> \[2\] Luo, Yuankai, Lei Shi, and Xiao-Ming Wu. "Classic GNNs are Strong Baselines: Reassessing GNNs for Node Classification." NeurIPS 2024\.
>
> \[3\] Shomer, Harry, et al. "Lpformer: An adaptive graph transformer for link prediction." ACM SIGKDD 2024\.
>
> \[4\] Mialon, Grégoire, et al. "Graphit: Encoding graph structure in transformers." arXiv preprint (2021).
>
> \[5\] Chen, Jinsong, et al. "NAGphormer: A Tokenized Graph Transformer for Node Classification in Large Graphs." ICLR 2023\.
>
> \[6\] Wu, Qitian, et al. "Nodeformer: A scalable graph structure learning transformer for node classification." NeurIPS 2022\.
>
> \[7\] Chen, Jinsong, et al. "Rethinking Tokenized Graph Transformers for Node Classification” NeurIPS 2025\.
>
> \[8\] Xing, Yujie, et al. "Unifying and Enhancing Graph Transformers via a Hierarchical Mask Framework” NeurIPS 2025\.
>
> \[9\] Kong, Kezhi, et al. "GOAT: A global transformer on large-scale graphs." ICML 2023\.
>
> \[10\] Zhang, Zaixi, et al. "Hierarchical graph transformer with adaptive node sampling.” NeurIPS 2022\.

---

### Author Response · Authors · 2025-12-03
**Discussion Period Summary (1/2)**

We sincerely thank all reviewers for their thoughtful and constructive feedback. We have incorporated all the suggested changes in our updated manuscript (highlighted in blue). Given the disruption to the discussion period, we are providing this short summary of the discussion period. We appreciate the AC’s time and effort in evaluating our submission under these unusual circumstances.

---
## Brief summary of our work:

Our paper investigates why graph transformers (GTs) underperform at the task of node classification and identifies eigenvector selection as the key bottleneck. We propose Broaden the Spectrum (BTS), a simple and principled method that selects Laplacian positional eigenvectors (LPEs) most beneficial for the task.

**Key contributions:**

1. **New theoretical framework and ESD:** We show that optimal attention matrices for node classification have a class-wise block structure. This provides us a metric, the Energy Spectral Density (ESD), to measure the usefulness of each eigenvector.
2. **The BTS algorithm:** We use ESD to rank and select the best eigenvectors. This leads to a lightweight, task-aware eigenvector selection method requiring no additional hyperparameters.
3. **Significant improvements to existing architectures:** Our empirical evaluations show that BTS consistently improves multiple existing graph transformers across a variety of benchmarks, and it enables simple architectures to state-of-the-art results.

We have also summarized the initial response of the reviewers in our “General Rebuttal Response” below.

---
## Addressing reviewer concerns:

Below we outline the key concerns we addressed and changes made to the paper:

### 1\. Additional baseline and benchmark comparisons (Reviewers wJCD, 3HAs):

- **Concern:** Recent SoTA transformer baselines (PolyFormer, SpecFormer, CoBFormer, Polynormer) were missing.

- **Our actions:**
  * We ran fair evaluations and added comparisons with four new baselines: PolyFormer, SpecFormer, Polynormer, CoBFormer.
  * We added results on filtered versions of the Chameleon and Squirrel dataset, as requested by 3HAs.
  * The updated results show that BTS variants of GT, NAGphormer, and GraphGPS remain highly competitive or state-of-the-art across datasets. For instance, GraphGPS moves from rank 6 → rank 1 when equipped with BTS.
  * We have updated Tables 1–2 and Figure 3 in the paper, plus added detailed tables in rebuttal.

### 2\. Performance under label-scarce settings (Reviewers m6Ny, wJCD):

- **Concern:** Since BTS relies on label information, does it remain effective when labels are scarce?

- **Our actions:**
  * Conducted new experiments with restricted label budgets (25%, 50%, 75%, 100%).
  * Conducted new few-shot experiments with 3, 5, 10 labels per class, as explicitly requested by wJCD.
  * Across all settings, GT-BTS consistently outperforms GT, demonstrating robustness even with very few labels. These results have been added to Appendix B, Figure 5, and the detailed tables provided in the rebuttal.

### 3\. Clarifications on block structure (Reviewer wJCD)

- **Concern:** Reviewer wJCD had conflicting intuition for what optimal attention matrices should be.

- **Our actions:** We clarified how their intuition was incomplete and also incorporated this clarification in the text. We also added more attention matrices to make our points more clear (Appendix E).

### 4\. Clarification on theoretical assumptions (Reviewer 3HAs)

- **Concern:** Reviewer 3HAs raised concerns about the data model in our theoretical framework.
- **Our response:** We clarified the plausibility of our assumptions. The reviewer was satisfied with our response, which was reflected by an increment of their score to 6 (as shown in their last response).

### 5\. Scalability and computation cost (Reviewers 3HAs, Ay4a)

- **Concern:** Reviewer 3HAs raised a concern that BTS depends on a full eigen-decomposition, potentially limiting its scalability to large graphs. Reviewer Ay4a also asked about the computation time for calculating the ESD.
- **Our actions:**
  * We clarified that BTS does *not* rely on full eigen-decomposition.
  * We clarified that the computational complexity of ESD calculation and BTS selection algorithm is negligible to the model run time.

### 6\. Graph classification (Reviewers m6Ny, 3HAs)

- **Concern:** Reviewers noted that our method only works for node classification.
- **Our response:** While this is a true limitation of our method, we note that there is significant precedence for focus on node classification (many published works that only focus on node classification are listed in our general responses).

---

> ### Author Response · Authors · 2025-12-03
> **Discussion Period Summary (2/2)**
>
> ### 7\. Positioning with respect to structural embeddings (Reviewer wJCD)
>
> - **Concern:** Reviewer wJCD expressed a concern about our phrasing suggesting that LPEs are the only approach for injecting positional information into graph transformers.
> - **Our action:** We updated our Introduction and the Related Works section to position our contribution within the broader landscape of different kinds of positional encoding methods.
>
> ### 8\. Miscellaneous
>
> There were a few misunderstandings and confusions expressed by reviewer Ay4a, that we have clarified in our rebuttal:
>
> - Novelty (they listed novelty as both a Strength and Weakness of our paper)
> - The “data-adaptive” nature of our method
> - Difference between BTS and other static position encodings
>
> ---
>
> We greatly appreciate the AC’s time, especially under the unusual review circumstances this year. We hope this summary, together with our full rebuttal and updated manuscript, helps provide a clear picture of the paper’s contributions and the extensive improvements made during the discussion period. Please let us know if any additional information would assist the evaluation.

---

### Meta-Review · Area_Chair_UviS · 2026-01-06

**Summary:**

This paper argues that graph transformers’ node-classification gap is partly due to simplistic heuristics for selecting Laplacian positional encodings, and proposes Broaden the Spectrum (BTS), which selects Laplacian eigenvectors by ranking their class-label energy spectral density computed from training labels. The authors provide a theoretical motivation via a simplified attention model in which an optimal attention matrix exhibits class-wise block structure, and they show that selecting eigenvectors with high label energy helps approximate such attention patterns. Empirically, BTS is plugged into several transformer backbones (GT, NAGphormer, GraphGPS) and reports large improvements on heterophilic, homophilic, and long-range benchmarks, especially when many eigenvectors are used alongside an adjusted encoder. Across reviews, the strengths were the clarity of the spectral diagnostic and the breadth of experiments, but concerns centered on whether the contribution is incremental (a label-guided top-k selection), and whether the required eigenvector computation plus extensive hyperparameter tuning limits scalability and reproducibility. Reviewers also raised questions about reliance on supervised labels (including low-label realism) and about how directly the paper’s block-structure theory and attention visualizations transfer to deep, softmax-based transformers with multi-head layers. After carefully reading the manuscript, appendices, and the full reviewer discussion, the AC finds that the remaining novelty and justification issues are not sufficiently resolved, and therefore rejection is recommended.

**Reviewer Concerns:**

The rebuttal convincingly addressed the request for stronger empirical baselines by adding comparisons to recent methods and by reporting results on filtered Chameleon/Squirrel splits. It also partially addressed label-dependence and scalability questions by adding label-scarcity and few-shot studies and clarifying that BTS can operate on a precomputed subset of eigenvectors with negligible overhead beyond the eigensolver. However, the key outstanding concern is that the core algorithm remains conceptually close to “project labels onto the spectrum then pick top-k,” and the theory relies on a highly simplified setting whose connection to actual transformer training dynamics is not fully convincing. In addition, doubts remain about practical reproducibility given the large tuning budget and about whether the attention-structure evidence is mechanistic rather than correlational, which together keep the overall confidence below the acceptance bar.

**Reviewer Scores:**

Reviewer wJCD would likely stay at 4 (marginally below threshold) because, despite acknowledging that baseline coverage and positioning were improved, they continued to flag unresolved issues around low-label realism and the interpretation of attention block structure. Reviewer m6Ny would likely move from 4 to 5 after seeing the added related-work discussion and the new label-scarcity experiments, while still viewing the contribution as somewhat task-limited. Reviewer Ay4a would likely remain at 4 or at most increase to 5 given the clarifications on “data-agnostic,” the compute-cost discussion, and the reproducibility scripts, but their novelty and experimental-search concerns would persist. Reviewer 3HAs explicitly indicated an increase to 6 in response to the added experiments and would likely keep a 6 with fuller discussion participation.

---

### Decision · Program_Chairs · 2026-01-26

Reject